# Don't Ignore the Tail: Decoupling top-$K$ Probabilities for Efficient Language Model Distillation

## Abstract

The core learning signal used in language model distillation is the standard Kullback-Leibler (KL) divergence between the distribution of the student and the teacher. Traditional KL divergence tends to be dominated by the teacher's highest-probability modes, thus diminishing the influence of less probable yet potentially informative components of the output distribution. We propose a new tail-aware divergence that decouples the contribution of the teacher model's top-$K$ predicted probabilities from those with lower probabilities, while maintaining the same computational profile as the KL Divergence. Our decoupled approach reduces the impact of the teacher modes and, consequently, increases the contribution of the tail of the distribution. Experimental results demonstrate that our modified distillation method yields competitive performance in both pre-training and supervised distillation of decoder models across various datasets. Furthermore, the distillation process is efficient and can be performed with a modest academic budget for large datasets, eliminating the need for industry-scale computing.[1]

## 1 Introduction

The rapid advancement in language models (LMs) has led to highly complex systems capable of performing state-of-the-art natural language processing (NLP) tasks. However, these models are often too computationally expensive and memory-intensive to be deployed on resource-constrained devices, such as edge devices, mobile phones, or low-latency systems. The gap is addressed by small language models, which can be further improved via knowledge distillation (KD) from larger models.

Most work on distilling generative language models focuses on supervised distillation, which aims to match the student's response to the teacher's response given a prompt (Gu et al. (2024), Agarwal et al. (2024)). These works typically assume the presence of an already pre-trained student, which might not always be the case. In contrast, works like DistilBERT (Sanh et al., 2019) train a student from scratch via pretraining distillation, and our work extends this technique to causal models. However, the training corpora for modern causal LMs are usually closed-source, which complicates the application of distillation approaches such as DistilBERT. However, applying pretraining distillation to modern causal LMs faces significant challenges: the training corpora are typically closed-source, and the models require substantial computational resources—often requiring tens to hundreds of billions of tokens when distilled on generic open-source corpora. This poses a significant computational challenge, especially in academic settings.

We distill various teacher models from different model families within a 1-week budget on a single H100 GPU, enabling the distillation of approximately 2 billion tokens for 1-billion-parameter student models, or more for smaller ones. We propose an algorithm that surpasses vanilla KD by decoupling the contribution of the teacher's top-$K$ probabilities to the KL divergence and demonstrate the method's effectiveness across different LMs. Despite the training budget constraint, our method produces competitive results with recent work, such as MiniPLM (Gu et al., 2025). Furthermore, when we use our supervised distillation method for mathematical reasoning, we achieve results comparable to SOTA scores on the same foundational models, with a GSM8K score of **36.8** for TinyLlama-1.1B and **56.0** for Llama2-7B after distillation.

---

[1]We used LLMs like Grammarly and ChatGPT-Plus to check grammar and spelling and to polish our work.

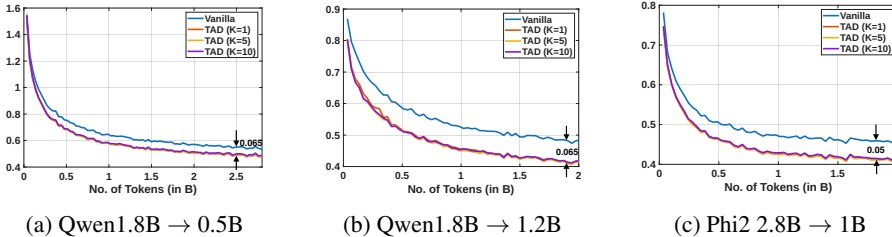

(a) Qwen1.8B → 0.5B      (b) Qwen1.8B → 1.2B      (c) Phi2 2.8B → 1B

Figure 1: KL divergence on the validation set of Regmix for vanilla KD vs TAD. The $x$ axis shows training progress in terms of the number of tokens, and the $y$ axis shows held-out KL between the student and teacher.

## 2    TAIL-AWARE DISTILLATION

If $\mathcal{P}$ is the simplex of token probabilities produced by a language model (e.g., $\mathcal{P}(S)$ for the student and $\mathcal{P}(T)$ for the teacher), then the standard distillation loss of a causal model has the following form for a sequence of length $N$,

$$\mathcal{L}_{KD} = \sum_{t=1}^{N} \mathcal{L}_{CLM}(t; \mathcal{P}^S) + \mathcal{D}_{KL}(t; \mathcal{P}^T, \mathcal{P}^S) \tag{1}$$

where $\mathcal{L}_{CLM}(t; \mathcal{P}^S)$ is the causal language modeling (CLM) loss of the student, and $\mathcal{D}_{KL}(t; \mathcal{P}^T, \mathcal{P}^S)$ is the KL divergence between the teacher and the student for the token $t$. In our method, we focus on the teacher's next-token probabilities when we input a sequence. With some abuse of notation, if $\overset{*}{p}{}_k^T = \max_{v \in \mathcal{V}}[\{p_1^T, p_2^T, \dots p_v^T \dots\} \setminus \{\overset{*}{p}{}_j^T\}_{j=1}^{k-1}]$ is the $k$th maximum of all the token probabilities for a vocabulary $\mathcal{V}$, we can split the KL divergence between the top-$K$ and the rest as,

$$\mathcal{D}_{KL}\left(\mathcal{P}^T \| \mathcal{P}^S\right) = \mathcal{D}_{KL}\left(p^T \| p^S\right)_{p^T \in \{\overset{*}{p}{}_k^T\}_{k=1}^K} + \alpha_K^T \mathcal{D}_{KL}\left(\tilde{p}^T \| \tilde{p}^S\right)_{p^T \notin \{\overset{*}{p}{}_k^T\}_{k=1}^K}$$
$$= \mathcal{D}_{KL_1} + \alpha_K^T \mathcal{D}_{KL_2} \tag{2}$$

Here $\{\overset{*}{p}{}_k^T\}_{k=1}^K$ is the set of top-$K$ teacher probabilities, and $\alpha_K^T = 1 - \sum_{k=1}^K \overset{*}{p}{}_k^T$ is the non-top-$K$ or the tail probability mass of the teacher. $\mathcal{D}_{KL_1}$ is the KL divergence associated with them (i.e., the modes), including a $(K+1)$st term for probabilities $1 - \sum_{k=1}^K \overset{*}{p}{}_k^T$ and $1 - \sum_{k=1}^K \overset{*}{p}{}_k^S$. Whereas, $\mathcal{D}_{KL_2}$ is the KL Divergence for the rest, i.e., the tail, involving $|\mathcal{V}| - K$ terms. The terms $\tilde{p}^T$ or $\tilde{p}^S$ in $\mathcal{D}_{KL_2}$ are the normalized teacher (or student) probabilities for the rest, i.e., $\tilde{p}^T = p^T / (1 - \sum_{k=1}^K \overset{*}{p}{}_k^T)$, since the sum of the non-top-$K$ probabilities is $1 - \sum_{k=1}^K \overset{*}{p}{}_k^T$. Note that even if the non-top-$K$ probabilities ($p^T \notin \{\overset{*}{p}{}_k^T\}_{k=1}^K$) are close to zero, their normalized values ($\tilde{p}^T$) are not. Therefore, $\mathcal{D}_{KL_2}$ is non-trivially different from zero. The detailed derivation is included in Appendix B.

Observe that if the probability distribution is skewed towards the modes, i.e., top-$K$ token probabilities and has a thin tail, $\sum_{k=1}^K \overset{*}{p}{}_k^T$ is very high, and the contribution of $\mathcal{D}_{KL_2}$ to the KL divergence is very low. To mitigate this, we can multiply the second term by a hyperparameter $\beta$, yielding the two-term loss $\mathcal{D}_{KL_1} + \beta \alpha_k^T \mathcal{D}_{KL_2}$. In this form, we recover the exact KL Divergence for $\beta = 1$, and the loss requires $\beta > 1$. Setting the value of $\beta$ becomes quite difficult, and the loss does not converge. We overcome this issue by sequence-level normalization. For the stochastic form of training, we use a mini-batch of sequences, and every token in a sequence has a different value of $\{p_1^T, p_2^T \dots, p_v^T\}$. If a sequence has $N$ tokens, we can normalize $\beta$ by the mean of $\alpha_K^T$ across all the tokens. Indexing the tokens with $t \in [N]$, the final loss for a token $t$ in the sequence takes the form,

$$\mathcal{L}_{DIV}(t; \mathcal{P}^T, \mathcal{P}^S) = D_{KL_1}(t) + \frac{\beta}{\frac{1}{N} \sum_{t=1}^N \alpha_k^T(t)} \alpha_k^T(t) D_{KL_2}(t) \tag{3}$$

This normalization makes the loss stable for nominal values of $\beta$, such as 1 or 2. This also preserves the overall shape of the teacher probability distribution, but only amplifies the tail's contribution to the KL divergence. Finally, we add the causal language modeling (CLM) loss of the student $\mathcal{L}_{CLM}(\mathcal{P}^S)$

for every token $t \in [N]$ to the divergence to constitute the final loss as,

$$\mathcal{L}_{TAD} = \sum_{t=1}^{N} \mathcal{L}_{CLM}(t; \mathcal{P}^S) + \mathcal{L}_{DIV}(t; \mathcal{P}^T, \mathcal{P}^S) \tag{4}$$

We refer to the original form of KD (Hinton et al., 2014) as Vanilla KD, which replaces $\mathcal{L}_{DIV}$ in Equation (4) with the KL divergence. When we train by optimizing $\mathcal{L}_{\text{DIV}}$ (see Section 3.2), the student attains a lower held-out KL than when trained by optimizing KL itself (Figure 1), even though KL is the evaluation metric. We also show the variation in tail probability mass ($\alpha_K^T$) with $K$ across different teachers in Figure 2.

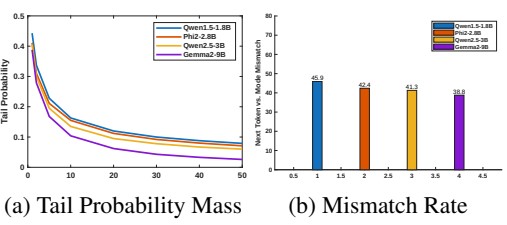

(a) Tail Probability Mass     (b) Mismatch Rate

Figure 2: Tail probability mass ($\alpha_K^T$) against $K$ for different teachers in the first, and the Next Token vs. Mode mismatch rate in percentage in the second plot, measured on the validation set of Regmix (see Section 3.2)

Our method is motivated by decoupled knowledge distillation (DKD; Zhao et al. (2022)), which was proposed for supervised classification with labeled datasets and improves accuracy on ImageNet and CIFAR-100. In contrast, language model pretraining distillation operates on unlabeled corpora, so the original DKD formulation is not well-suited to this setting. While one might treat the next token as a target label, this creates a fundamental mismatch: in classification, the target class is, by definition, correct. However, since most LMs' pretraining corpora are undisclosed and we distill using a generic corpus, the teacher's most probable token (i.e., $\arg\max_{v \in \mathcal{V}} p_v^T$) may differ from the ground-truth next token. When we study this discrepancy on the validation set of our dataset (see Section 3.2), we observe a mismatch rate ranging from 39% to 46%, depending on the teacher, with larger teachers having lower mismatch rates (Figure 2). This mismatch creates conflicting signals between the dataset labels and teacher predictions. We therefore introduce TAD: a rank-based Top-$K$ vs. tail decoupling using a probability-mass-normalized tail KL divergence that preserves the teacher's distributional information. TAD is not a variant of DKD: DKD's decoupling is label-anchored (target vs. non-target), while TAD's is rank-anchored (Top-$K$ vs tail) and label-free. Two examples with identical values of $p^S$ and $p^T$ yield the same TAD losses, but their DKD losses can be different if their labels differ.

## 2.1 Gradient Analysis

For a token $t$ in a sequence $X$ of length $N$, the KL Divergence loss is $\mathcal{L}_{KLD} = \sum_{i=1}^{|\mathcal{V}|} p_i^T \log(p_i^T / p_i^S)$, where the probabilities $p_i$ are typically produced by the softmax of the logit $z_i$ of the final layer, the gradient has the following form. For the sake of simplicity, we omit the index $t$ from the equations.

$$\frac{\partial \mathcal{L}_{KLD}}{\partial z_i} = p_i^S - p_i^T \tag{5}$$

Since the top-$K$ probabilities of the teacher, denoted $\mathring{p}_k^{*T}$, are much larger than the tail probabilities (i.e., $\mathring{p}_k^{*T} \gg p_i^T$ for $k \in [K]$, $i \in [\mathcal{V} \setminus K]$), the gradients w.r.t the logits of the top-$K$ tokens are much greater than the those of the tail tokens' logits. This forces the student model to focus primarily on the top-$K$ tokens, pushing the sum of the student's top-$K$ probabilities close to 1, i.e., $\sum_{k=1}^{K} \mathring{p}_k^{*S} \approx 1$.

For Tail-aware KD, the gradient of the loss w.r.t the logits of the top-$K$ probabilities remains the same as Equation (5). However, for the tail logits ($z_i : i \in [\mathcal{V} \setminus K]$), it has the form

$$\frac{\partial \mathcal{L}_{DIV}}{\partial z_i} = p_i^S - p_i^T + \left(\beta(X) - 1\right) \left( p_i^S \cdot \frac{1 - \sum_{k=1}^{K} \mathring{p}_k^{*T}}{1 - \sum_{k=1}^{K} \mathring{p}_k^{*S}} - p_i^T \right) \tag{6}$$

where $\beta(X) = \beta / (\frac{1}{N} \sum_{t=1}^{N} \alpha_k^T(t))$ is defined in Equation (3) and is specific to the sequence $X$, and $\mathring{p}_k^{*S}$ are the student probabilities corresponding to the tokens of top-$K$ teacher probabilities. We typically set $\beta \geq 1$, and $\beta(X)$ has probability terms in the denominator, making $\beta(X) > 1$. When $\sum_{k=1}^{K} \mathring{p}_k^{*S} \approx 1$, the second term of $\nabla_{z_i} \mathcal{L}_{DIV}$ (Equation (6)) increases the relative weight of tail gradients, causing the tail probability of the student to rise, ensuring that $\sum_{k=1}^{K} \mathring{p}_k^{*S} < 1$.

This mechanism ensures that the tail probability of the student will rise with each gradient step as long as the top-$K$ probability of the student is more than the teacher's, i.e., $\sum_{k=1}^{K} \overset{*}{p}_k^S \geq \sum_{k=1}^{K} \overset{*}{p}_k^T$. In this case, the gradient satisfies: $\nabla_{z_i} \mathcal{L}_{DIV} \geq \beta(X)(p_i^S - p_i^T)$, which is stronger than the standard KL gradient. Once the top-$K$ probability mass of the student matches the teacher's, i.e., $\sum_{k=1}^{K} \overset{*}{p}_k^S \approx \sum_{k=1}^{K} \overset{*}{p}_k^T$, the gradient compensation stops. At this point, the $\nabla_{z_i} \mathcal{L}_{DIV} \approx \beta(X)(p_i^S - p_i^T)$. The fixed point of the gradient lies at $p_i^S = p_i^T$, same as Vanilla KD, and therefore converges to the same solution. By this stage, the student has already acquired a sufficient mass in the tail probabilities and has begun to generalize beyond the top-$K$ tokens. On the other hand, if $\sum_{k=1}^{K} \overset{*}{p}_k^S < \sum_{k=1}^{K} \overset{*}{p}_k^T$, the strong gradient of top-$K$ tokens will drive up the top-$K$ probability mass of the student. This way, Tail-aware KD enables a better learning of the teacher probabilities across the entire vocabulary. The full derivation is included in the Appendix C.

## 3 EXPERIMENTAL DETAILS

We distill models of varying sizes, ranging from Qwen1.5 (1.8B) to Gemma-2 (9 B). We do not have access to (or require) the pretraining corpus of any of these models. MiniPLM was trained on the Pile dataset (Gao et al., 2020), an extensive 825 GB collection that is no longer available due to copyrighted content. We instead use a small 20GB subsample[2] of the Regmix dataset (Liu et al., 2024b), containing a total of 5B tokens, that can be processed using our limited compute setting. Regmix replicates the Pile, but without copyrighted components.

We only perform pretraining distillation in our experiments, and **no fine-tuning** is done on any labeled dataset for the student models. Unless mentioned otherwise, we use a temperature of 1 and a context size of 2048 for all our distillation experiments. The training details, including the exact architecture of the students, hardware, and hyperparameters, are detailed in Appendix A.

### 3.1 EVALUATION

We evaluate the models on eight datasets for few-shot performance, as in Gu et al. (2025), using the standard LM evaluation harness (Gao et al., 2024) from Huggingface (Wolf et al., 2019), and then report the average score across all datasets.

### 3.2 PRETRAINING DISTILLATION FROM SCRATCH

We follow Sanh et al. (2019) in using the teacher's weights to initialize the student models, by initializing the student's attention layers with the teacher's attention weights, truncated to the student's hidden dimension for each head. The MLP layers are randomly initialized.

#### 3.2.1 BENCHMARKING WITH QWEN

We begin our experiments by distilling the Qwen1.5-1.8B model to benchmark our method against the recently published MiniPLM (Gu et al., 2025). It is a data-centric distillation method that utilizes the teacher to identify suitable samples for training the student, but it cannot perform supervised distillation. Table 1 also reports the results of Sequence-KD (Kim & Rush, 2016) and MiniLLM (Gu et al., 2024) for comparison, quoted from the MiniPLM article. Sequence-KD fine-tunes the student on teacher-generated sequences. MiniLLM records the student's generated output in response to a prompt and uses a reward maximization algorithm similar to PPO (Schulman et al., 2017). DistilLM (Ko et al., 2024) is a similar algorithm to MiniLM, producing results similar to MiniLM while reducing execution time; therefore, it is not mentioned separately. These experiments are expensive (costs reported in Table 3), and reproducing them on billions of tokens was infeasible with our resources.

Consistent with MiniPLM, we distill the model to two students with 1.2B and 0.5B parameters, corresponding to approximately 1B and 475M active (non-embedding) parameters, respectively. We use only 2B tokens to distill the 1.2B model and 2.8B tokens for the 0.5B model — as much as we could train on an H100 GPU within a week. Note that MiniPLM trains the student on

---

[2]`https://huggingface.co/datasets/sail/regmix-data-sample`

| Teacher/Student | | HS | WG | OBQA | ARC-E | ARC-C | PIQA | SIQA | Story | Avg | $\overline{\text{Rel}}$ |
|---|---|---|---|---|---|---|---|---|---|---|---|
| | CLM (no KD) | 39.4 | 51.8 | 28.4 | 46.0 | 25.7 | 67.0 | 39.5 | 62.2 | 45.0 | −0.7 |
| | Vanilla KD | 40.7 | 53.2 | 29.8 | 46.1 | 25.5 | 67.3 | 39.2 | 63.5 | 45.6 | |
| | Seq-KD | 38.5 | 51.9 | 29.2 | 46.5 | 25.1 | 66.3 | 39.0 | 61.0 | 44.7 | −0.9 |
| Qn1.5 | MiniLLM | 36.1 | 51.2 | 28.5 | 44.1 | 25.3 | 65.8 | 37.9 | 61.4 | 43.8 | −1.9 |
| 1.8B | MiniPLM | **42.8** | 53.3 | 31.0 | 46.8 | 26.9 | **68.3** | 39.8 | **64.0** | 46.6 | +1.0 |
| ↓ | TAD ($K=1$) | 42.3 | 53.8 | 30.5 | 52.0 | 27.0 | 67.3 | **41.2** | 63.9 | 47.2 | +1.2 |
| 1.2B | TAD ($K=5$) | 42.9 | 53.9 | **31.7** | 52.3 | 27.0 | 68.1 | 41.1 | 63.5 | 47.6 | +2.1 |
| | TAD ($K=10$) | **43.0** | **55.2** | 31.5 | **53.1** | 27.1 | **68.2** | 40.9 | 63.6 | **47.8** | **+2.2** |
| | TAD ($K=20$) | 42.8 | 54.7 | 30.9 | 52.7 | **27.6** | 68.1 | 41.0 | 63.5 | 47.7 | +2.0 |
| | CLM (no KD) | 35.8 | 51.0 | 30.2 | 41.7 | 24.4 | 65.4 | 38.2 | 61.4 | 43.6 | −0.5 |
| | Vanilla KD | 37.0 | 51.7 | 29.4 | 45.1 | 24.2 | 65.8 | 38.0 | 61.6 | 44.1 | |
| | Seq-KD | 34.9 | 50.7 | 28.6 | 42.7 | 23.6 | 65.0 | 38.4 | 58.9 | 42.8 | −1.3 |
| Qn1.5 | MiniLLM | 33.0 | 51.2 | 27.5 | 42.1 | 24.2 | 62.3 | 37.3 | 60.2 | 42.3 | −1.9 |
| 1.8B | MiniPLM | **39.0** | **52.2** | 30.2 | **45.8** | 24.9 | **67.0** | 39 | **62.2** | 45.0 | +1.0 |
| ↓ | TAD ($K=1$) | 38.0 | 51.7 | 30.5 | 45.9 | 25.7 | 66.7 | 39.4 | 61.7 | 45.0 | +1.1 |
| 0.5B | TAD ($K=5$) | 38.2 | 52.0 | 31.0 | 45.8 | 25.8 | 66.9 | 39.7 | 61.7 | 45.1 | +1.3 |
| | TAD ($K=10$) | 38.4 | **52.1** | **31.1** | **46.0** | 25.9 | **67.3** | **39.8** | **62.2** | **45.4** | **+1.5** |
| | TAD ($K=20$) | 38.2 | 50.3 | 31.0 | 45.2 | 25.3 | 66.1 | 39.6 | 62.1 | 44.7 | +0.9 |

Table 1: Results for Tail-aware distillation for $\beta = 2$ over Qwen1.5-1.8B ("Qn"), for a 1.2B and 0.5B student model. The best performance for each column, and any value within $0.4$ of it, is highlighted. CLM stands for pre-training the model with only the CLM loss, without distillation. The average relative change for the best-case TAD ($K = 10$) is 50% to 120% better than MiniPLM.

anywhere from 25 to 50B tokens and draws inference on the teacher over 100B tokens, a much larger computational budget than in our case. We perform the distillation for $K \in \{1, 5, 10, 20\}$, following the experimental settings used in prior work on top-$K$ based methods (Lapin et al., 2016; Kool et al., 2019). Results improve until K= 10, beyond which there is not much benefit. For the optimal setting of $K = 10$, we conduct a sensitivity analysis over $\beta \in \{0.5, 1, 2, 5, 10\}$, with results presented in Table 2. Performance peaks around $\beta = 2$, with a smooth degradation on either side up to $\beta = 1$, indicating robustness to this hyperparameter. However, for $\beta < 1$, the performance might degrade fast as $\beta(X) > 1$ is no longer guaranteed (Equation (6)).

| | $\beta$ | 0.5 | 1 | 2 | 5 | 10 |
|---|---|---|---|---|---|---|
| 1.2B | Avg | 47.0 | 47.6 | **47.8** | 47.7 | 47.6 |
| | $\overline{\text{Rel}}$ | +1.4 | +2.0 | **+2.2** | +2.1 | +2.0 |
| 0.5B | Avg | 45.0 | 45.1 | **45.4** | 45.1 | 44.9 |
| | $\overline{\text{Rel}}$ | +1.0 | +1.2 | **+1.5** | +1.2 | +1.0 |

Table 2: Parameter sensitivity of $\beta$ for the distillation of Qwen 1.8B for $K = 10$

For the 1.2B student model, Tail-aware KD consistently outperforms MiniPLM's average score by a substantial margin across all values of $K$. For the smaller 0.5B student, the performance gap narrows, though Tail-aware KD still maintains an edge. A breakdown by task shows that TAD outperforms MiniPLM across more challenging benchmarks, such as ARC-Challenge and OpenBookQA. In contrast, MiniPLM exhibits slight gains on easier tasks, such as ARC-Easy and Story. Since the easier tasks inherently yield higher accuracy, the averages tend to be skewed towards them. To provide a more granular evaluation, we compute the symmetric relative change in accuracy with respect to Vanilla KD, following Törnqvist et al. (1985). The relative change is defined as Rel = $100 \cdot \log(\text{Acc}/\text{Acc}_{\text{Vanilla}})$, where Acc is the accuracy of the method under comparison (e.g., MiniPLM or TAD). We report the average relative change across all tasks as $\overline{\text{Rel}}$ in Table 1. The difference between MiniPLM and TAD becomes more prominent in the relative measure.

MiniPLM approximates reverse-KL–style distillation via data selection: the teacher scores the corpus, selects suitable samples, and the student is then trained on those samples. However, to sample an $\delta$ fraction of the corpus, it takes $1/\delta$ times as many forward passes through the teacher as backpropagations through the student, which is a significant overhead. When we compute the FLOPs for all the methods to train on 1M tokens, MiniPLM has **33**% to **50**% higher FLOP count due to the overhead (Table 3), while TAD has a similar FLOP count to Vanilla KD. The authors of MiniPLM

| Teacher/Student | | HS | WG | OBQA | ARC-E | ARC-C | PIQA | SIQA | Story | Avg | $\overline{\text{Rel}}$ | F-ECE ↓ |
|---|---|---|---|---|---|---|---|---|---|---|---|---|
| | CLM (no KD) | 38.2 | 51.1 | 27.4 | 51.2 | 24.1 | 66.3 | 40.8 | 63.1 | 45.3 | −4.3 | 1.57 |
| Phi2 | CLM (Mat.) | 40.2 | 51.9 | 28.6 | 52.3 | 24.8 | 67.6 | 41.7 | 64.7 | 46.5 | −2.4 | 1.50 |
| 2.8B | Vanilla KD | 43.6 | 53.5 | 33.0 | 57.3 | 30.0 | 68.0 | 43.2 | 64.3 | 49.1 | | 1.45 |
| ↓ | MiniPLM | 43.7 | 52.5 | 30.6 | 57.1 | 29.9 | 68.1 | 43.8 | 64.3 | 48.8 | −0.4 | 1.62 |
| 1B | RKL | 42.3 | 54.1 | 31.6 | **58.0** | 28.7 | 68.2 | 43.8 | **64.9** | 49.0 | −0.4 | 1.77 |
| | TAD ($K = 1$) | 45.2 | 55.3 | 34.0 | 58.0 | 30.7 | 68.3 | 44.4 | **64.9** | 50.1 | +0.9 | 1.19 |
| | TAD ($K = 5$) | 45.5 | 55.6 | **34.6** | 58.1 | 31.0 | 68.8 | **44.5** | 64.7 | **50.3** | +1.2 | 1.29 |
| | TAD ($K = 10$) | **45.6** | 56.0 | 34.0 | **58.3** | **31.1** | 68.8 | 43.8 | 64.7 | 50.3 | +1.1 | 1.37 |
| | TAD ($K = 20$) | 45.3 | **56.4** | 33.5 | 57.6 | 31.0 | **69.0** | 43.8 | 64.7 | 50.2 | +1.0 | 1.42 |
| | CLM (no KD) | 36.2 | 53.0 | 26.4 | 46.6 | 25.9 | 61.6 | 35.7 | 58.9 | 43.0 | −1.9 | 1.49 |
| Qn2.5 | CLM (Mat.) | 38.1 | 53.9 | 27.6 | 47.6 | 26.6 | 62.8 | 36.5 | 60.4 | 44.2 | −0.7 | 1.41 |
| 3B | Vanilla KD | 38.0 | 53.4 | 26.8 | 50.6 | 27.4 | 64.0 | 38.8 | 60.4 | 44.9 | | 1.42 |
| ↓ | MiniPLM | 37.3 | 53.4 | **29.2** | 49.4 | 25.3 | **64.7** | 38.6 | **61.4** | 44.9 | +0.0 | 1.45 |
| 1B | RKL | 38.9 | 53.7 | 28.2 | 50.7 | 27.6 | 63.8 | 39.0 | 61.4 | 45.4 | +0.6 | 1.99 |
| | TAD ($K = 1$) | 39.9 | 54.3 | 27.5 | 52.1 | 27.8 | **64.9** | **39.7** | 60.9 | 45.9 | +1.0 | 1.29 |
| | TAD ($K = 5$) | 39.9 | 53.5 | 27.9 | **53.4** | 27.9 | 64.9 | 39.2 | 61.0 | 46.0 | +1.1 | 1.30 |
| | TAD ($K = 10$) | **40.6** | 54.5 | **29.6** | 52.0 | 28.4 | 64.8 | 39.3 | **61.5** | **46.3** | +1.6 | 1.32 |
| | TAD ($K = 20$) | 40.5 | 54.5 | 29.2 | 51.8 | **29.1** | 64.3 | 39.6 | 61.2 | 46.2 | +1.6 | 1.37 |
| | CLM (no KD) | 37.4 | 49.2 | 27.2 | 49.0 | 25.1 | 65.4 | 38.9 | 60.7 | 44.1 | −1.7 | 1.43 |
| Gem2 | CLM (Mat.) | 39.4 | 50.0 | 28.4 | 50.1 | 25.8 | 66.7 | 39.8 | 62.2 | 45.3 | −0.4 | 1.41 |
| 9B | Vanilla KD | 40.3 | 51.3 | 27.8 | 53.0 | 26.1 | 66.9 | 39.2 | 61.9 | 45.8 | | 1.27 |
| ↓ | MiniPLM | 37.5 | 51.9 | 27.2 | 49.5 | 26.0 | 66.6 | 39.0 | 61.9 | 46.0 | −0.8 | 1.56 |
| 2B | RKL | 39.4 | 52.0 | 28.1 | 53.4 | 26.3 | 66.8 | **40.1** | 62.5 | 46.1 | +0.2 | 1.80 |
| | TAD ($K = 1$) | 41.0 | 52.1 | 28.4 | 54.0 | 26.4 | **67.6** | 39.3 | 61.9 | 46.3 | +0.5 | 1.04 |
| | TAD ($K = 5$) | **41.3** | 52.7 | 28.5 | 54.2 | 26.5 | 67.3 | 39.7 | 62.2 | 46.5 | +0.6 | 1.11 |
| | TAD ($K = 10$) | 41.2 | **53.7** | 30.0 | **54.5** | **26.8** | 67.1 | **40.1** | 62.8 | **47.0** | +1.3 | 1.17 |
| | TAD ($K = 20$) | 40.9 | 52.8 | **30.0** | **54.5** | 26.3 | 66.9 | 39.7 | 62.4 | 46.7 | +1.0 | 1.20 |

Table 4: Pretraining distillation of various teachers to students with ∼1B active parameters on 2 billion tokens from Regmix. CLM (no KD) refers to pretraining with only CLM loss, without distillation with the same number of tokens (2B), where CLM (Mat.) refers to computation-matched pretraining, matched to the same FLOPs as training of TAD. The last column "F-ECE" shows the calibration error of the models, measured using Full-ECE, with the lower being better.

treat the teacher-scoring overhead as offline pre-processing, as they use the same teacher for all their students. However, a practitioner might want to try different teachers to optimize a small LM rather than relying on a single teacher, or even use a multi-teacher approach for optimal performance, as in Wu et al. (2021). Unlike any divergence-based method, MiniPLM cannot be applied to such practical scenarios without significant modification. Finally, MiniPLM is not necessarily competitive with our approach, and its selected samples could, in principle, be used with our tail-aware divergence as the distillation loss. However, we exclude such combinations from the scope of this work.

### 3.2.2 DISTILLING LARGER MODELS

We further distill a series of larger models in Table 4, namely Phi-2 (Javaheripi et al., 2023), Qwen2.5-3B (Yang et al., 2024), and Gemma2-9B (Team et al., 2024), with parameter size ranging from 2.8B to 9B. We choose teacher checkpoints only with pretraining to ablate the effect of instruction tuning on distillation. The student's architectures are selected to have the same dimensions as the teacher's, but with fewer layers and smaller intermediate sizes. For medium-sized models like Phi-2 or Qwen2.5-3B, the student has half the teacher layers, whereas for Gemma2-9B, the student has a third of the teacher's layers. The student embeddings are initialized from the teacher embeddings and remain frozen thereafter, resulting in approximately 1B active

| # P(M) | Vanilla | MiniPLM | TAD | MiniLLM | Seq-KD |
|---|---|---|---|---|---|
| **1.2B** | 9.2 | 12.4 | 9.3 | 39.0 | 65.0 |
| **0.5B** | 6.4 | 9.7 | 6.5 | 21.8 | 43.2 |

Table 3: PetaFLOPs for the distillation of Qwen-1.5-1.8B (Section 3.2.1) on a subset of 1M tokens from the Regmix dataset. TAD has a similar PFLOP to Vanilla KD, while MiniPLM is higher than both. The methods involving sequence generation (SeqKD or MiniLLM) are too expensive to scale to billions of tokens.

| No. of Tokens | Phi2 2.8 → 1B | | | Qn2.5 3B → 1B | | | Gemma 9B → 2B | | |
|---|---|---|---|---|---|---|---|---|---|
| | 10B | 100B | 1T | 10B | 100B | 1T | 10B | 100B | 1T |
| Vanilla KD (KL) | 2.80 | 2.77 | 2.76 | 3.18 | 3.09 | 3.05 | 3.15 | 3.00 | 2.93 |
| Vanilla KD (RKL) | 2.91 | 2.86 | 2.84 | 3.26 | 3.15 | 3.11 | 3.23 | 3.08 | 3.01 |
| TAD ($K = 10$) | 2.78 | 2.73 | 2.71 | 3.04 | 2.92 | 2.87 | 3.08 | 2.94 | 2.88 |

Table 5: Validation loss predictions for three distillation methods—Vanilla KD with forward and reverse KL divergence and Tail-aware Distillation (TAD, $K = 10$), fit with the scaling law of (Hoffmann et al., 2022). TAD is projected to achieve the lowest loss even when scaled to 1T training tokens.

parameters per student. For example, Gemma2-9B has around 900M embedding parameters due to its large vocabulary size (256K), so the 2B student has only 1.1B active parameters. We also add cosine loss between the student and the teacher hidden states to Equation (4), similar to DistilBERT (Sanh et al., 2019). Finally, we add MiniPLM experiments on the same training dataset in Table 4. Due to computational constraints, we do not train a reference model from scratch; instead, we use OPT-125M (Zhang et al., 2022) as a reference model for all the teachers. We used a difference-sampling ratio of $\delta = 0.5$, the same as in the MiniPLM experiments.

When we measure the distillation cost in PetaFLOPs on a small training subset containing 1M tokens as in the last section, MiniPLM takes **50**% more FLOPs as Vanilla KD for the distillation of Phi2 (**18**.4 vs. **12**.4) or Qwen2.5-3B (**22**.2 vs. **15**.2), and **67**% more for Gemma2 (**52**.0 vs. **31**.4). At the same time, TAD has a similar FLOP count to Vanilla KD. For the entire distillation, both the Vanilla KD and TAD exceed $10^{19}$ FLOPs per billion tokens for teachers with 3B or more parameters. To put this into perspective, the pretraining distillation of the older models, such as MBART-Large (610M params, Tang et al. (2020)), consumes at most $10^{17}$ FLOPs overall (Dasgupta & Cohn, 2025). We do not present any baseline other than Vanilla KD and MiniPLM, as we already demonstrated the high computational cost of MiniLLM and Seq-KD in the previous section (Table 3).

The students receive no fine-tuning after distillation, and we evaluate them on the same few-shot tasks as before. MiniPLM did not outperform Vanilla KD, and on Phi-2 it was worse (Table 4). Adding the cosine loss on hidden states improved both Vanilla KD and TAD. As formulated, MiniPLM (a data-selection method) does not incorporate such internal-state losses, which reduces its competitiveness relative to Section 3.2.1. To ensure parity, we also report reverse KL (RKL) with the same cosine loss on the hidden states (Table 4). RKL is slightly better than vanilla KD but remains inferior to TAD. For TAD, the performance improved up to $K = 5$ or 10, beyond which we did not see any significant gain (Table 4). We also evaluate the loss using Equation (1) on the Regmix validation set and extrapolate it to 1T training tokens with the scaling law in Hoffmann et al. (2022). The projected losses in Table 5 show that TAD surpasses the other methods when distilled with large token budgets.

### 3.2.3 CALIBRATION ERROR

We evaluate model calibration using Expected Calibration Error (ECE) (Table 4). Specifically, we adopt the Full-ECE metric from (Liu et al., 2024a), which is tailored to language models with large vocabularies and measures calibration over the entire predictive distribution, rather than the standard ECE from (Guo et al., 2017), which focuses only on the argmax prediction and is more appropriate for classification settings. We found that TAD has a slightly lower Full-ECE than Vanilla KD. However, the ECE increases with $K$ for all the cases. The reverse KL has the worst ECE of all.

### 3.2.4 SELECTION OF $K$

Across experiments with Qwen1.5-1.8B (Section 3.2.1) and with the larger teacher models, we observe that performance peaks at $K = 5$ or 10 and then declines. In natural language, the next-token probabilities are approximately Zipfian, and the teacher's tail mass $\alpha_K^T(t) = 1 - \sum_{k=1}^{K} \mathring{p}_k^T(t)$ decay sharply beyond $K \gtrsim 5$–10 (see Figure 2). Even after normalizing the tail term in $\mathcal{L}_{DIV}$ by the sequence mean $\bar{\alpha}_K^T = \frac{1}{N} \sum_{t=1}^{N} \alpha_K^T(t)$ of the tail probability mass, many low-entropy tokens still satisfy $\alpha_K^T(t) \to 0$ as $K$ grows. Instead, the contribution of high-entropy (noisier) tokens increases with $K$. Consequently, we observe no material gains beyond $K \approx 5$–10.

| Model | #Tkn | HS | WG | OBQA | ARC-E | ARC-C | PIQA | SIQA | Story | Avg |
|---|---|---|---|---|---|---|---|---|---|---|
| **TinyLlama(TL)−1.1B** | 1T | 53.6 | 56.8 | 32.2 | 61.2 | 30.1 | 70.8 | 41.2 | 68.1 | 51.8 |
| CLM (no KD) | +2B | 53.8 | 57.1 | 32.6 | 61.8 | 30.4 | 71.1 | 41.4 | 68.5 | 52.1 |
| **Phi3 4B** ↓ **TL** — Vanilla KD | +2B | 54.1 | 58.9 | 33.5 | 63.2 | 31.6 | 71.2 | 44.2 | 68.7 | 53.2 |
| TAD ($K = 1$) | | 55.1 | 60.0 | 34.8 | **64.2** | **33.1** | 71.3 | 44.8 | 69.1 | 54.0 |
| TAD ($K = 5$) | +2B | **55.5** | **60.2** | **35.2** | 63.6 | 32.6 | 71.4 | **44.9** | **69.4** | **54.1** |
| TAD ($K = 10$) | | 54.6 | 60.0 | 34.6 | 63.2 | 32.4 | 71.8 | 44.6 | 68.9 | 53.8 |
| TAD ($K = 20$) | | 54.8 | 59.9 | 33.8 | 62.5 | 32.0 | **72.1** | 44.2 | 68.7 | 53.5 |
| **TinyLlama(TL)−1.1B** | 2T | 55.2 | 58.9 | 33.4 | 61.3 | 30.7 | 71.4 | 42.1 | 68.9 | 52.7 |

Table 6: Continued pretraining for the distillation of Phi-3 models to TinyLlama-1B. We use the TinyLlama-1B checkpoint, pretrained on 1T tokens, as the student and distill it on an additional 2B tokens from the Regmix corpus. The distilled students outperform the 2T checkpoint of TinyLlama, by training on $500\times$ less tokens.

### 3.3 CONTINUED PRETRAINING

Although we demonstrated that our distillation algorithm works across various sizes of teacher models, it is not possible to create student models from scratch with only 2B tokens to achieve state-of-the-art performance. In this section, we start from an already pretrained student model, TinyLlama-1.1B (Zhang et al., 2024), specifically its 1T checkpoint, and focus on distilling it from Phi-3 (Abdin et al., 2024), a much stronger model. In the first set of experiments, we distill the students on the same 2B tokens from the Regmix dataset. Here, we do not use any teacher model internals, nor do we freeze the student embeddings. As in the previous sections, no fine-tuning is performed on the students after distillation. The distilled students outperform the 2T checkpoint of TinyLlama (Table 6), trained with another 1T tokens ($500\times$) from the base model.

#### 3.3.1 MATHEMATICAL REASONING

In this section, we distill TinyLlama-1.1B using Phi3-Mini as the teacher on the OpenWebMath (OWM) corpus (Paster et al., 2023), which primarily consists of mathematical articles. The Distillation is performed on 2.5 billion tokens from the token, and the 2.5T TinyLlama-1.1B checkpoint is used as the base model. Evaluation is performed on eight tasks using the standard setting of Mathematical evaluation harness,[3] namely GSM8K, MATH, SVAMP, ASDiv, MAWPS, Tabmwp (TAB), MathQA (MQA), and SAT (Table 7). We employ a few-shot chain-of-thought approach (Wei et al., 2022) for evaluation and then measure the average score across the tasks.

Tiny-Llama performs poorly in mathematical reasoning tasks. After distillation, we observe approximately 2 times better performance on tasks such as MAWPS, MATH, and ASDiv, and 3.5 times better on GSM8K. Furthermore, the distilled students with TAD outperform Llama3.2-1B, which is pretrained with a far higher number of tokens (9T), whereas Vanilla KD falls short. These

---

[3] https://github.com/ZubinGou/math-evaluation-harness

| Model | Data (#Tkns) | GSM8K | MATH | SVAMP | ASDiv | MAWPS | TAB | MQA | SAT | Avg |
|---|---|---|---|---|---|---|---|---|---|---|
| **TinyLlama(TL)−1.1B** | Web (2.5T) | 2.0 | 2.6 | 9.5 | 16.3 | 20.1 | 12.7 | 12.8 | 15.6 | 11.4 |
| CLM (no KD) | + OWM(2.5B) | 3.9 | 3.8 | 17.9 | 29.7 | 39.5 | 12.2 | 10.8 | 15.6 | 16.7 |
| **Phi3 4B** ↓ **TL** — Vanilla KD | + OWM(2.5B) | 6.1 | 4.2 | 21.1 | 33.5 | 41.5 | 15.5 | 11.2 | 16.7 | 18.7 |
| MiniPLM | | 3.3 | 3.4 | 13.4 | 27.3 | 34.0 | 10.8 | 10.5 | 12.5 | 14.4 |
| TAD ($K = 1$) | | 6.1 | **6.2** | **22.1** | 33.1 | 41.5 | 14.0 | 11.3 | 21.9 | 19.5 |
| TAD ($K = 5$) | + OWM(2.5B) | **7.1** | 4.8 | 19.2 | **35.9** | **46.7** | 15.9 | 10.0 | 22.6 | 20.3 |
| TAD ($K = 10$) | | 6.4 | 4.6 | 19.7 | 33.0 | 42.7 | 12.9 | 9.3 | **37.5** | **20.7** |
| TAD ($K = 20$) | | 6.5 | 3.8 | 18.2 | 31.7 | 40.9 | 13.7 | 9.0 | 31.2 | 19.4 |
| **Gemma3−1B−PT** | Web (2T) | 2.1 | 2.2 | 12.8 | 17.1 | 22.4 | 11.1 | **14.5** | 15.6 | 12.2 |
| **Llama3.2−1.2B−PT** | Web (9T) | 6.5 | 4.2 | **21.7** | 35.7 | 44.2 | **21.1** | 13.2 | 6.2 | 19.1 |

Table 7: Adaptation to mathematical reasoning via pretraining distillation of Phi-3 into TinyLlama-1B ("TL") on the OpenWebMath (OWM) corpus. The distilled students with TAD outperform pretrained 1B Gemma3 and Llama3.2 models in terms of average score.

| Model | Data (#tokens) | GSM8K | MATH | SVAMP | ASDiv | MAWPS | TAB | MQA | SAT | Avg. |
|---|---|---|---|---|---|---|---|---|---|---|
| TinyLlama(TL)−1.1B | Web (2.5T) | 2.0 | 2.6 | 9.5 | 16.3 | 20.1 | 12.7 | 12.8 | 15.6 | 11.4 |
| CLM + SFT | +OWM(2.5B) +ORCAMEL | 19.6 | 4.0 | 49.4 | 58.8 | 74.3 | 21.8 | 18.0 | 28.1 | 34.3 |
| Phi3 4B ↓ TL   Vanilla KD | +OWM(2.5B) +ORCAMEL | 30.8 | 6.8 | 64.6 | 62.5 | 80.7 | 20.1 | 16.7 | 27.5 | 38.7 |
| TAD ($K=1$) | +OWM (2.5B) +ORCAMEL | **36.8** | 6.8 | **67.8** | 67.9 | 81.7 | 25.4 | 16.3 | 28.1 | 41.4 |
| TAD ($K=5$) | | 33.2 | 7.4 | 65.4 | **68.7** | **85.6** | 27.6 | 17.9 | **34.4** | **42.5** |
| TAD ($K=10$) | | 30.1 | 9.0 | 65.7 | 68.4 | 85.4 | 24.1 | 18.2 | 29.8 | 41.3 |
| TAD ($K=20$) | | 28.2 | 7.2 | 66.2 | 68.2 | 84.2 | 24.6 | 17.1 | 25.0 | 40.1 |
| Rho−1−Math(1.1B) | +OWM (30B) † | 36.3 | **13.4** | 52.6 | 66.5 | 83.6 | **29.5** | **32.1** | 18.5 | 41.5 |
| Llama2−7B | Web (2T) | 14.2 | 3.6 | 39.1 | 51.6 | 63.6 | 30.9 | 12.5 | 32.8 | 31.4 |
| CLM + SFT | +OWM(2.5B) +ORCAMEL | 22.0 | 4.2 | 47.7 | 56.3 | 72.3 | 37.7 | 23.0 | 28.1 | 36.4 |
| Phi3 14B ↓ L2   Vanilla KD | +OWM(2.5B) +ORCAMEL | 50.5 | 8.1 | 75.3 | 74.4 | 90.5 | 29.7 | 37.2 | 34.4 | 50.0 |
| TAD ($K=1$) | +OWM (2.5B) +ORCAMEL | **56.0** | 10.2 | 77.2 | **77.1** | 91.8 | 39.8 | 39.2 | 40.6 | 54.0 |
| TAD ($K=5$) | | 51.6 | 9.2 | 76.7 | 75.4 | 91.2 | 38.7 | **40.5** | 37.5 | 52.6 |
| TAD ($K=10$) | | 51.4 | 8.4 | 76.6 | 75.5 | 90.6 | 38.7 | 39.2 | 44.4 | 53.1 |
| TAD ($K=20$) | | 52.8 | 8.0 | **77.6** | 76.9 | **92.4** | 39.2 | 39.0 | **46.9** | **54.1** |
| Llemma−7B | +ProofPile(0.2T) | 39.7 | **15.4** | 56.9 | 67.7 | 83.3 | **47.0** | **40.9** | 44.0 | 49.4 |
| WizardMath−7B | +RL with Evol Instruct | 46.6 | 7.0 | 56.8 | 65.2 | 81.1 | 35.0 | 20.3 | 23.1 | 41.9 |
| Orca2−7B | +SFT (ORCA) + KTO | 40.0 | 6.2 | 70.2 | 67.0 | 87.5 | 30.4 | 31.6 | 28.1 | 45.1 |

† Trained with special Rho loss to eliminate the noisy tokens.

Table 8: Supervised distillation for mathematical reasoning, showing distillation of Phi3-4B into TinyLlama-1.1B ("TL") and Phi3-14B into Llama2-7B on ORCAMEL, alongside GPT4-generated solutions. TAD for TinyLlama is $2.5\times$ computationally cheaper than Rho-1 and $9\times$ cheaper for Llama2-7B than Llemma-7B (see Appendix A.1), which is the best model created from Llama2-7B.

experiments demonstrate that a seemingly weak student model (e.g., TinyLlama) can be made competitive in a specific domain through distillation from an expert teacher. For MiniPLM, we choose Galactica-125m (Taylor et al., 2022) as the reference model, since it is pretrained on scientific datasets including mathematics, and uses a difference sampling ratio of $\delta = 0.5$. MiniPLM completely fails for domain-specific distillation, with an average score worse than pretraining without distillation (CLM in Table 7).

## 3.4 SUPERVISED DISTILLATION

For our final experiment, we perform supervised distillation for mathematical reasoning using instructions generated from GPT-4 (Table 8). We combine a 200K dataset from Microsoft-ORCA (Mitra et al., 2024) with a 50K dataset from Camel-AI (Li et al., 2023), both of which contain answers generated by GPT-4 in response to mathematical questions, and refer to the combined dataset as ORCAMEL. Unlike many mathematical instruction datasets, e.g., Yu et al. (2023), which use the training responses from GSM8K (Cobbe et al., 2021) or MATH (Lewkowycz et al., 2022), our training dataset contains only their input prompts, making the results more generalizable. Furthermore, we do not use any modifications of the original question as an intermediate step, such as backward questions in Yu et al. (2023) or Evol-Instructions in Luo et al. (2023), which might yield additional gains.

We perform our distillation on two pairs of teacher and student: (1) Phi3-4B to TinyLlama, and (2) Phi3-14B to Llama2-7B (Touvron et al., 2023). We do not fine-tune the teachers on the dataset and assume them to be sufficiently capable in mathematical reasoning to produce supervision signals. For every pair of teacher and student, our distillation is performed in two stages,

1. Pretraining distillation on 2.5B tokens from the OWM corpus ($\beta = 2.0$)

2. Three epochs of distillation on the ORCAMEL dataset for the same teacher–student pair.

We also add a baseline by fine-tuning TinyLlama on the ORCAMEL dataset, after pretraining it on the same 2.5B OWM tokens without any distillation. The performance of the distilled models is comparable to that of Rho-1 (Lin et al., 2024). Rho-1 is created by continuing TinyLlama's

pretraining on 30B tokens from the OWM corpus, using reducible holdout (Rho) loss selection (Mindermann et al., 2022) to eliminate noisy tokens, achieving SOTA results on mathematical tasks with models of around 1B parameters. The distilled Llama2-7B outperforms SOTA models for Maths inference built using Llama-2 as the base model, such as Llemma-7B (Azerbayev et al., 2023), Orca-2 (Mitra et al., 2024), or Wizard-Math (Luo et al., 2023), and we generated their results using the same Mathematical evaluation harness. Further, our method has a much lower compute budget than the next-best model, Llama-7 B, as explained in Appendix A.1. Although unsupervised corpora for pretraining are unlimited, supervised datasets are always limited. It is better to use them with a teacher's supervision for optimal performance, rather than merely fine-tuning the student on them.

## 4 RELATED WORK

Most of the work in KD for LLMs focuses on task-specific knowledge transfer via instruction prompts, following Sequence-KD (Kim & Rush, 2016), where the teacher generates a sequence-specific prompt, and the student is fine-tuned on that sequence. Recently, there has been a surge in reinforcement learning-based policy optimization for distillation, like MiniLLM and Agarwal et al. (2024). However, these methods involve generating sequences from the student during training, which can be expensive for large datasets. Recently, DistilLM (Ko et al., 2024) addressed this issue by implementing an efficient generation scheduler. Overall, these on-policy methods are limited to small datasets; for example, both DistilLM and MiniLLM use the DollyEval dataset, which contains 15,000 data points. They cannot be applied to large-scale datasets larger than 200K, which is standard for distillation for Summarization or Translation (Shleifer & Rush (2020), Agarwal et al. (2024)).

When it comes to large-scale pretraining distillation to prepare the student from scratch, there is work on encoder-only models, such as DistilBERT (Sanh et al., 2019) or MiniLM (Wang et al., 2020). Work like Shleifer & Rush (2020) extends it to encoder–decoder models for generative tasks such as summarization or machine translation. However, most pretraining distillation in causal models, such as distilling Gemma2 models from Gemini (Team et al., 2024) or work like Muralidharan et al. (2024), still follows logit matching with minimal modification. MiniPLM is the only work we found that attempts distillation without logit matching.

Works like MiniPLM, MiniLLM, or On-policy KD of Agarwal et al. (2024) uses the reverse KL divergence instead of the forward one. However, the mode-seeking behavior of reverse KLD will suppress the contribution of words other than the one with the maximum probability. For task-specific distillation, where we match the conditional teacher probability ($\mathbb{P}[y|x]$) on the output sequence $y$ given a prompt input $x$, mode-seeking might be beneficial. However, for pretraining distillation on the entire input $x$, we match $\mathbb{P}[x]$ for every token. The teacher's probability distribution will contain multiple dominant modes, and focusing solely on the maximum will limit the transfer of dark knowledge. Furthermore, a strong correlation exists between KD and reward maximization for aligning language models, as established in the derivation of MiniPLM. Wang et al. (2023) shows that preference alignment using the reverse KL divergence lowers the diversity of a model's generated sequence, and the same will be true for KD as well.

## 5 CONCLUSION

Here, we present a novel distillation algorithm for language models that extends the commonly used KL divergence, and we demonstrate its competitiveness through extensive experiments. Works such as Sequence-KD and MiniLLM are not well-suited to pretraining on large-scale datasets. MiniPLM performs poorly for domain-specific distillation and cannot be directly applied to supervised tasks. In contrast, our method applies to both pretraining and supervised distillation, and it is significantly cheaper in the latter because it requires neither teacher decoding (as in Seq-KD) nor student generation (as in MiniLLM or DistilLM (Ko et al., 2024)). Consequently, TAD has a computational burden comparable to Vanilla KD, enabling large-scale pretraining distillation within a limited GPU budget. Finally, we show that it can be used to train competitive models for mathematical reasoning using publicly available datasets. Taken together with its low computational requirements, TAD provides a compelling and versatile distillation method for causal LMs.

## 6 ETHICS STATEMENT

Critical ethical considerations in training language models include licensing terms of the pre-training data; evaluation and mitigation of model bias with respect to a variety of protected attributes of both users and target referents; and AI safety guardrails over the final model to reduce toxic/harmful outputs. As this paper centers on a novel knowledge distillation method and all experiments use widely used language models and open-source datasets, there are no new dimensions to these concerns. We do, however, concede that KD can amplify existing model biases to some degree (Ahn et al., 2022), that it is possible to mitigate teacher model biases as part of the KD process (Blakeney et al., 2021), and that there is value in quantifying this effect for our method. We consider this to be orthogonal to this work, however.

## 7 REPRODUCIBILITY

We have attached a few code samples as supplementary material. The teacher models and the datasets are all open-source and available on huggingface (Wolf et al., 2019). The data preprocessing step involves standard random sampling without replacement from datasets like Regmix (Section 3.2) or Open-Web-Math (Section 3.3.1).

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

| Teacher | #P(M) | $|\mathcal{V}|$ | $d_S$ | $L_S$ | $n_H$ | $d_H$ | $d_{FFN}$ |
|---------|-------|-----------------|-------|-------|-------|-------|-----------|
| Qwen1.5-1.8B | 1.2B | 151,936 | 1,536 | 24 | 16 | 96 | 4,224 |
| Qwen1.5-1.8B | 0.5B | 151,936 | 1,024 | 24 | 16 | 64 | 2,816 |
| Phi2-2.8B | 1.1B | 52,000 | 2,560 | 16 | 32 | 80 | 5,120 |
| Qwen2.5-3B | 1.2B | 151,936 | 2,048 | 18 | 16 | 128 | 7,680 |
| Gemma2-9B | 2B | 256,000 | 3,584 | 14 | 16 | 224 | 4,096 |

Table 9: The architectures of different students used in distillation for pretraining from scratch. $|\mathcal{V}|$ is the vocabulary size, $d_S$ for the hidden size of the student, $L_S$ for the number of layers, and $n_H$ for the number of heads, $d_H$ for the dimension of each head, and $d_{FFN}$ for the intermediate size.

## A  EXPERIMENTAL DETAIL

The architectures of different students for the pretraining from scratch are listed in Table 9. All students have approximately 1B active parameters, except for the 0.5B student of Qwen, which has approximately $475M$ active parameters. The architectures of the students of Qwen$1.5 - 1.8$B are kept the same as in the MiniPLM paper (Gu et al., 2025).

The experiments are divided into two major parts: pretraining distillation from scratch, and continued pretraining. For pretraining distillation from scratch, we distilled the Qwen1.5, Phi2, and Qwen2.5 models on a single H100 GPU for a week, whereas we used 2 H100 GPUs for distilling the Gemma2-9B model. We used flash attention (Dao et al., 2022) whenever possible to speed up the computation, except for Gemma2. We used Adam optimizer (Kingma & Ba, 2014) with a learning rate of $\eta = 1e-4$ and a weight decay of $\lambda_d = 0.1$ for all the experiments. We used a batch size of 128 for all the experiments.

For the continued pretraining distillation of Tiny-Llama, we used the Adam optimizer (Kingma & Ba, 2014) with a learning rate of $\eta = 3e - 5$ and a weight decay of $\lambda_d = 0.1$ for all experiments. All experiments used a batch size of 128 and were conducted on a single NVIDIA H100 GPU. Supervised distillation is performed with a batch size of 32, $\eta = 1e - 5$, $\lambda_d = 0.1$, and a context size of 2048.

### A.1  COST OF SUPERVISED DISTILLATION

We conduct a comparative cost analysis of GPU hours required to produce state-of-the-art mathematical reasoning, starting with foundational models such as TinyLlama-1.1B and Llama2-7B. Models like Llemma or Rho-1 are trained using industrial resources. Rho-1 is trained for approximately 10 hours on a 32-GPU H100 stack, requiring a total of 320 GPU hours. The best model built on Llama-7B is Llemma, which was trained on A100 GPUs for 23,000 GPU hours. Even though it uses different hardware, we can draw an equivalence using the GPU hours the 7B model in Lin et al. (2024) takes to train on H100. It required 18 hours to train on 15 billion tokens using 32 H100 GPUs. Using their configuration setting, Llemma-7B will take 7,680 GPU hours to train on a single H100. This provides a reasonable estimate, since A100s are approximately a third slower than H100 GPUs for training ($23K \approx 3 \times 7,680$). Our two-stage method requires approximately 130 hours on a single H100 GPU for TinyLlama and 420 hours on two H100 GPUs (totaling 840 hours) for Llama-2, which is substantially cheaper than the existing methods.

## B  SPLITTING OF THE KL DIVERGENCE INTO TOP-$K$ AND TAIL

Here, we show the derivation of Equation (2). Like we defined before, $\overset{*}{p}_k^T = \max_v [\{p_1^T, p_2^T \ldots p_v^T\} \setminus \{\overset{*}{p}_j^T\}_{j=1}^{k-1}]$ is the $k$th maximum of all the token probabilities for a vocabulary size $v$, and $\overset{*}{p}_k^S$ is the corresponding student probability of the same word. The sum of top-K probability of the teacher is $\sum_{k=1}^{K} \overset{*}{p}_k^T$. The normalized teacher (or student) probability, by the factor $1 - \sum_{k=1}^{K} \overset{*}{p}_k^T$, is defined as,

$$\tilde{p}_T = \frac{p^T}{1 - \sum_{k=1}^{K} \overset{*}{p}_k^T} \qquad \tilde{p}_S = \frac{p^S}{1 - \sum_{k=1}^{K} \overset{*}{p}_k^S} \tag{7}$$

It can be easily seen that for the non-top-K probabilities, $\tilde{p}^T$ sums to 1, i.e. $\sum_{p^T \notin \{\overset{*}{p}_k^T\}_{k=1}^K} \tilde{p}^T = 1$. Now, we split the KL divergence between the top-K probability and the rest, as follows,

$$
\begin{aligned}
&\mathcal{D}_{KL}\left(\mathcal{P}^T \| \mathcal{P}^S\right) \\
&= \sum_{p^T \in \{\overset{*}{p}_k^T\}_{k=1}^K} p^T \log \frac{p^T}{p^S} + \sum_{p^T \notin \{\overset{*}{p}_k^T\}_{k=1}^K} p^T \log \frac{p^T}{p^S} \\
&= \sum_{p^T \in \{\overset{*}{p}_k^T\}_{k=1}^K} p^T \log \frac{p^T}{p^S} + \left(1 - \sum_{k=1}^K \overset{*}{p}_k^T\right) \sum_{p^T \notin \{\overset{*}{p}_k^T\}_{k=1}^K} \frac{p^T}{\left(1 - \sum_{k=1}^K \overset{*}{p}_k^T\right)} \log \frac{p^T}{p^S} \\
&= \sum_{p^T \in \{\overset{*}{p}_k^T\}_{k=1}^K} p^T \log \frac{p^T}{p^S} + \left(1 - \sum_{k=1}^K \overset{*}{p}_k^T\right) \sum_{p^T \notin \{\overset{*}{p}_k^T\}_{k=1}^K} \tilde{p}^T \log \frac{\tilde{p}^T \left(1 - \sum_{k=1}^K \overset{*}{p}_k^T\right)}{\tilde{p}^S \left(1 - \sum_{k=1}^K \overset{*}{p}_k^S\right)} \\
&= \sum_{p^T \in \{\overset{*}{p}_k^T\}_{k=1}^K} p^T \log \frac{p^T}{p^S} + \left(1 - \sum_{k=1}^K \overset{*}{p}_k^T\right) \sum_{p^T \notin \{\overset{*}{p}_k^T\}_{k=1}^K} \tilde{p}^T \log \frac{1 - \sum_{k=1}^K \overset{*}{p}_k^T}{1 - \sum_{k=1}^K \overset{*}{p}_k^S} \\
&\qquad\qquad + \left(1 - \sum_{k=1}^K \overset{*}{p}_k^T\right) \sum_{p^T \notin \{\overset{*}{p}_k^T\}_{k=1}^K} \tilde{p}^T \log \frac{\tilde{p}^T}{\tilde{p}^S} \\
&= \sum_{p^T \in \{\overset{*}{p}_k^T\}_{k=1}^K} p^T \log \frac{p^T}{p^S} + \left(1 - \sum_{k=1}^K \overset{*}{p}_k^T\right) \log \frac{1 - \sum_{k=1}^K \overset{*}{p}_k^T}{1 - \sum_{k=1}^K \overset{*}{p}_k^S} \underbrace{\left(\sum_{p^T \notin \{\overset{*}{p}_k^T\}_{k=1}^K} \tilde{p}^T\right)}_{1} \\
&\qquad\qquad + \left(1 - \sum_{k=1}^K \overset{*}{p}_k^T\right) \mathcal{D}_{KL}\left(\tilde{p}^T \| \tilde{p}^S\right)_{p^T \notin \{\overset{*}{p}_k^T\}_{k=1}^K} \\
&= \mathcal{D}_{KL}\left(p^T \| p^S\right)_{p^T \in \{\overset{*}{p}_k^T\}_{k=1}^K} + \left(1 - \sum_{k=1}^K \overset{*}{p}_k^T\right) \mathcal{D}_{KL}\left(\tilde{p}^T \| \tilde{p}^S\right)_{p^T \notin \{\overset{*}{p}_k^T\}_{k=1}^K} \\
&= \mathcal{D}_{KL_1} + \left(1 - \sum_{k=1}^K \overset{*}{p}_k^T\right) \mathcal{D}_{KL_2}
\end{aligned}
\tag{8}
$$

## C  DERIVATION OF THE GRADIENT

Here we present an elaborated derivation of the gradients. The derivations follow the material in the appendix of Anshumann et al. (2025). If $p_i = \exp(z_i)/\sum_{i=1}^{|\mathcal{V}|} \exp(z_i)$ is the softmax probability for a logit $z_i$ for a vocabulary $\mathcal{V}$, then the gradient of $p_k$ is (from (Iwana et al., 2019)):

$$
\frac{\partial p_j}{\partial z_i} = p_j \left(\mathbb{1}_{[i=j]} - p_i\right)
\tag{9}
$$

Now, given a vocabulary $\mathcal{V}$, the KL Divergence loss between the teacher probabilities of the teacher $(p_i^T)$ and the student $(p_i^S)$ is:

$$
\mathcal{L}_{KLD} = \sum_{i=1}^{|\mathcal{V}|} p_i^T \log(p_i^T / p_i^S)
\tag{10}
$$

It can be derived that,

$$\frac{\partial \mathcal{L}_{KLD}}{\partial z_i} = -\sum_{j=1}^{|\mathcal{V}|} \frac{p_j^T}{p_j^S} \frac{\partial p_j^S}{\partial z_i} = -\sum_{j=1}^{|\mathcal{V}|} p_j^T \left( \mathbb{1}_{[i=j]} - p_i^S \right)$$

$$= p_i^S \cdot \left( \sum_{j=1}^{|\mathcal{V}|} p_j^T \right) - \sum_{j=1}^{|\mathcal{V}|} p_j^T \mathbb{1}_{[i=j]}$$

$$= p_i^S - p_i^T \tag{11}$$

Now, we can show that $\mathcal{D}_{KL_1}$ has $K+1$ terms when we consider top-$K$ probabilities, with the first $K$ being ($i \in [K]$)

$$L_{1:K} = \sum_{k=1}^{K} \overset{*}{p}_k^T \log \frac{\overset{*}{p}_k^T}{\overset{*}{p}_k^S}$$

where $\overset{*}{p}_k^S$ are the student probabilities corresponding to the top-$K$ tokens, i.e. tokens for which the teacher probabilities are maximum. The derivative of $L_{1:K}$ w.r.t a logit $z_i$ is

$$\frac{\partial L_{1:K}}{\partial z_i} = p_i^S \cdot \left( \sum_{k=1}^{K} \overset{*}{p}_k^T \right) - \sum_{k=1}^{K} \overset{*}{p}_k^T \mathbb{1}_{[i=k]} \tag{12}$$

Now for $i \in [\mathcal{V} \setminus K]$, the indicator function $\mathbb{1}_{[i=k]}$ is never one. Therefore, the gradient of $L_{1:K}$ has the following forms for two different cases, as:

$$\frac{\partial L_{1:K}}{\partial z_i} = \begin{cases} p_i^S \cdot (\sum_{k=1}^{K} \overset{*}{p}_i^T) - p_i^T & i \in [K] \\ p_i^S \cdot (\sum_{k=1}^{K} \overset{*}{p}_i^T) & i \in [\mathcal{V} \setminus K] \end{cases}$$

Please note that the top $K$ probabilities do not sum to one. The last term $L_{K+1}$ can be expressed as:

$$L_{K+1} = \left( 1 - \sum_{i=1}^{K} \overset{*}{p}_k^T \right) \log \frac{1 - \sum_{i=1}^{K} \overset{*}{p}_k^T}{1 - \sum_{i=1}^{K} \overset{*}{p}_i^S} = -\left( 1 - \sum_{k=1}^{K} \overset{*}{p}_k^T \right) \cdot \log \left( 1 - \sum_{k=1}^{K} \overset{*}{p}_k^S \right) + \mathbf{C}$$

where C is a constant. The derivative of the last term, using the derivative of $p_k^S$ from Equation (9) is:

$$\frac{\partial L_{K+1}}{\partial z_i} = \frac{1 - \sum_{k=1}^{K} \overset{*}{p}_k^T}{1 - \sum_{k=1}^{K} \overset{*}{p}_k^S} \cdot \sum_{k=1}^{K} \frac{\partial \overset{*}{p}_k^S}{\partial z_i} = \frac{1 - \sum_{k=1}^{K} \overset{*}{p}_k^T}{1 - \sum_{k=1}^{K} \overset{*}{p}_k^S} \cdot \sum_{k=1}^{K} \overset{*}{p}_k^S \left( \mathbb{1}_{[i=k]} - p_i^S \right)$$

Again, for $i \in [\mathcal{V} \setminus K]$, the indicator function $\mathbb{1}_{[i=k]}$ is never one. Therefore,

$$\frac{\partial L_{K+1}}{\partial z_i} = \begin{cases} p_i^S \cdot \left( 1 - \sum_{k=1}^{K} \overset{*}{p}_k^T \right) & i \in [K] \\ -p_i^S \cdot \left( \frac{1 - \sum_{k=1}^{K} \overset{*}{p}_k^T}{1 - \sum_{k=1}^{K} \overset{*}{p}_k^S} \right) \sum_{k=1}^{K} \overset{*}{p}_k^T & i \in [\mathcal{V} \setminus \mathcal{K}] \end{cases} \tag{13}$$

Combining the gradients of $L_{1:K}$ and $L_{K+1}$, since $\mathcal{D}_{KL_1} = L_{1:K} + L_{K+1}$

$$\frac{\partial \mathcal{D}_{KL_1}}{\partial z_i} = \begin{cases} p_i^S - p_i^T & i \in [K] \\ p_i^S \cdot \left( \frac{\sum_{k=1}^{K} \overset{*}{p}_k^T - \sum_{k=1}^{K} \overset{*}{p}_k^S}{1 - \sum_{k=1}^{K} \overset{*}{p}_k^S} \right) & i \in [\mathcal{V} \setminus \mathcal{K}] \end{cases} \tag{14}$$

Therefore, the gradients of the logits corresponding to the tokens of top-$K$ teacher probabilities remain the same, while the gradients of the logits corresponding to the rest of the tokens change. The second term $\mathcal{D}_{KL_2}$ solely depends on the logits of the rest of the tokens.

$$\mathcal{D}_{KL_2} = \sum_{i \in \mathcal{V} \setminus K} \tilde{p}_i^T \log \frac{\tilde{p}_i^T}{\tilde{p}_i^S} \tag{15}$$

where we can generate $\tilde{p}_i^S$ directly from $z_i$ as $\tilde{p}_i^S = \frac{\exp z_i}{\sum_{k \in \mathcal{V} \setminus K} \exp z_k}$. Also, $\tilde{p}_i^T$ comes from a similar softmax, but is constant. Therefore,

$$\frac{\partial \mathcal{D}_{KL_2}}{\partial z_i} = \begin{cases} 0 & i \in [K] \\ \tilde{p}_i^S - \tilde{p}_i^T & i \in [\mathcal{V} \setminus \mathcal{K}] \end{cases}$$

The gradients of the logits of the top-$K$ tokens are zero for $\mathcal{D}_{KL_2}$; only their gradient for $\mathcal{D}_{KL_1}$ is non-zero (Equation (14)). And as a result, their gradient is the same as that for ordinary KL Divergence (Equation (11)). Therefore, TAD does **not** change the gradient of the logits of the top-$K$ tokens.

As for the logits of the non-top-$K$ tokens, their gradient for $\mathcal{D}_{KL_2}$ can be written as,

$$\frac{\partial \mathcal{D}_{KL_2}}{\partial z_i} = \frac{p_i^S}{1 - \sum_{k=1}^{K} \overset{*}{p}_k^S} - \frac{p_i^T}{1 - \sum_{k=1}^{K} \overset{*}{p}_k^T} \tag{16}$$

since $\tilde{p}_i^T$ and $\tilde{p}_i^S$ can also be defined as Equation (7).

Therefore,

$$\left(1 - \sum_{k=1}^{K} \overset{*}{p}_k^T\right) \frac{\partial \mathcal{D}_{KL_2}}{\partial z_i} = p_i^S \cdot \frac{1 - \sum_{k=1}^{K} \overset{*}{p}_k^T}{1 - \sum_{k=1}^{K} \overset{*}{p}_k^S} - p_i^T \tag{17}$$

Combining the derivative of $\mathcal{D}_{KL_2}$ from (Equation (14)) for the tail logits, i.e., for $i \in [\mathcal{V} \setminus \mathcal{K}]$, it can easily be checked that

$$\frac{\partial \mathcal{D}_{KL_1}}{\partial z_i} + \left(1 - \sum_{k=1}^{K} \overset{*}{p}_k^T\right) \frac{\partial \mathcal{D}_{KL_2}}{\partial z_i}$$

$$= \left(\frac{p_i^S \cdot \sum_{k=1}^{K} \overset{*}{p}_k^T - p_i^S \cdot \sum_{k=1}^{K} \overset{*}{p}_k^S}{1 - \sum_{k=1}^{K} \overset{*}{p}_k^S}\right) + \left(\frac{p_i^S - p_i^S \cdot \sum_{k=1}^{K} \overset{*}{p}_k^T}{1 - \sum_{k=1}^{K} \overset{*}{p}_k^S}\right) - p_i^T$$

$$= p_i^S - p_i^T$$

Since $\mathcal{L}_{KLD} = \mathcal{D}_{KL_1} + \left(1 - \sum_{k=1}^{K} \overset{*}{p}_k^T\right) \mathcal{D}_{KL_2}$, their gradients are the same. Now, for TAD, the divergence is: $\mathcal{L}_{DIV} = \mathcal{D}_{KL_1} + \beta(X) \left(1 - \sum_{k=1}^{K} \overset{*}{p}_k^T\right) \mathcal{D}_{KL_2}$, where $\beta(X) = \beta / (\frac{1}{N} \sum_{t=1}^{N} (1 - \sum_{k=1}^{K} \overset{*}{p}_k^T(t)))$, where $t$ is the index of a token in a sequence $X$ containing a total of $N$ tokens. This also means,

$$\mathcal{L}_{DIV} = \mathcal{D}_{KL_1} + \left(1 - \sum_{k=1}^{K} \overset{*}{p}_k^T\right) \mathcal{D}_{KL_2} + (\beta(X) - 1) \left(1 - \sum_{k=1}^{K} \overset{*}{p}_k^T\right) \mathcal{D}_{KL_2}$$

$$= \mathcal{L}_{KLD} + (\beta(X) - 1) \left(1 - \sum_{k=1}^{K} \overset{*}{p}_k^T\right) \mathcal{D}_{KL_2}$$

Using Equation (17), the gradient of $\mathcal{L}_{DIV}$ has the following form for the logits $z_i$ for the tail tokens ($i \in [\mathcal{V} \setminus K]$)

$$\frac{\partial \mathcal{L}_{DIV}}{\partial z_i} = \frac{\partial \mathcal{L}_{KLD}}{\partial z_i} + (\beta(X) - 1) \left(1 - \sum_{k=1}^{K} \overset{*}{p}_k^T\right) \frac{\partial \mathcal{D}_{KL_2}}{\partial z_i}$$

$$= p_i^S - p_i^T + (\beta(X) - 1) \left(p_i^S \cdot \frac{1 - \sum_{k=1}^{K} \overset{*}{p}_k^T}{1 - \sum_{k=1}^{K} \overset{*}{p}_k^S} - p_i^T\right)$$

For the logits of the top-$K$ tokens, $\frac{\partial \mathcal{D}_{KL_2}}{\partial z_i} = 0$, and therefore, their gradients are the same as those of Vanilla KD. This completes the derivation of the gradient of $\mathcal{L}_{DIV}$.