# OpenReview forum: "Don't Ignore the Tail: Decoupling top-$K$ Probabilities for Efficient Language Model Distillation"
_ICLR.cc/2026/Conference — Submitted to ICLR 2026_

### Official Review · Reviewer_gFdZ · 2025-10-27

**Soundness:** 3
**Presentation:** 3
**Contribution:** 3
**Rating:** 6
**Confidence:** 5

**Summary:**

This paper introduces a tail-aware knowledge distillation (KD) method for language models, termed TAD. The approach decouples the top-$K$ probabilities of the teacher distribution from the tail (remaining probabilities) in the KL divergence loss, thereby increasing the gradient signal for lower-probability outputs (the "tail") during distillation. This decoupling allows the student model to better capture information carried by less probable teacher outputs, overcoming the well-known issue where vanilla KD is dominated by teacher models. The authors present mathematical derivations, a thorough gradient analysis, and comprehensive empirical results across several pretraining and supervised distillation tasks. Notably, their approach achieves competitive or superior accuracy on multiple benchmarks, including mathematical reasoning tasks, all within modest computational constraints.

**Strengths:**

1. This paper clearly points out a key weakness of standard knowledge distillation -- that the loss is mainly affected by the teacher's top tokens. It then suggests a clear and logical way to fix this problem by dividing the teacher's output distribution based on token ranks. The mathematical explanations show a solid understanding of their arguments.

2. The paper addresses compute constraints and provides FLOP cost comparisons, establishing that TAD is practical for academic-scale training and accessible without industry-level GPUs.

**Weaknesses:**

1. The paper's treatment of the tail amplification factor $\beta$ is largely empirical and lacks theoretical grounding. Section 2 does not provide formal analysis of convergence, stability, or statistical effects from scaling the tail term, even though $\beta$ is described as sensitive. Without clearer justification, it remains uncertain when amplifying low-probability tokens improves generalization versus merely amplifying noise.

2. Although the evaluations are broad among LLMs and mathematical domains, most experiments are focused on either standard few-shot evaluation tasks or mathematical reasoning datasets. The absence of tasks in other domains -- such as dialogue modeling, summarization after supervised training.

3. The codebase is not released at time of review. While the paper claims code will be made available upon acceptance, currently, replication depends on implementation from scratch.

**Questions:**

See weaknesses.

---

> ### Author Response · Authors · 2025-12-03
> **Rebuttal**
>
> **Weakness:**
> 1. We appreciate the reviewer’s concern and agree that blindly amplifying low-probability tokens could, in principle, amplify noise. Our treatment of $\beta$ combines (i) a local, gradient-level analysis in Sec. 2 and (ii) systematic empirical evidence, rather than a full convergence proof.
>
> First, TAD is constructed so that $\beta$ does not change the optimum: the loss remains a sum of KL terms between the teacher and student (for both head and renormalized tail), so the unique fixed point is still $p^S = p^T$ for any $\beta > 0$. In Sec.2 we derive the gradient of the TAD loss and show that the tail term contributes a factor proportional to $\beta/ \alpha_K^T \cdot \big(\tilde{p}^S - \tilde{p}^T\big)$, where $\tilde{p}^S,\tilde{p}^T$ are the normalized tail distributions. Thus $\beta$ simply rescales the relative strength of a KL-consistent tail correction; it cannot introduce a new minimizer, and it does not drive the student toward arbitrary “noise” distributions.
>
> Second, we analyze the *direction* of the update of the head mass of the student ($\Delta P^S_{\text{head}}$ in Rebuttal for w25M) and show that, for moderate $\beta$, TAD amplifies the tail only when the student is over-concentrated on the head compared to the teacher. In other words, the tail term acts as a regularizer that counterbalances over-confident heads rather than indiscriminately pushing probability mass into rare tokens. This is reflected in our empirical results: across all teacher–student pairs, we observe *both* improved accuracy and consistently lower Full-ECE compared to vanilla KD and reverse KL, indicating that tail amplification with $\beta \approx 2$ improves calibration rather than merely injecting noise.
>
> Finally, we do not treat $\beta$ as a highly sensitive knob: our ablations show a wide plateau for $\beta \in [1,5]$, and we use the same default $\beta=2$ across all models and domains without per-task tuning. A full statistical generalization and stability analysis of this family of KL-compatible reweightings is, in our view, an interesting open theoretical problem, but beyond the scope of this empirical distillation study.
>
> 2. We agree that it is valuable to understand how TAD behaves beyond the standard few-shot and mathematical reasoning benchmarks. Two considerations drive our choice of evaluation settings: (i) measuring generalization rather than in-distribution test accuracy, and (ii) focusing on domains where tail behavior is particularly important.
>
> First, we use widely adopted few-shot benchmarks to evaluate the students’ generalization capability across a range of tasks. In much of the prior work on dialogue modeling and summarization, the student is distilled on a single dataset (e.g., XSum or SQuAD) and evaluated on the test split of that same dataset. This primarily measures in-distribution performance and does not adequately probe out-of-distribution behavior or tail calibration.
>
> In contrast, for mathematical reasoning, we deliberately use a mismatch between distillation and evaluation: we distill from GPT-generated solutions on a training corpus, but evaluate on few-shot or generative math benchmarks via a standard mathematical evaluation harness. This setup better captures how well the student extrapolates beyond the distillation data and is more sensitive to the distribution of probability mass between the head and tail, which is precisely what TAD targets.
>
> Second, we view math and few-shot benchmarks as a stress test for the TAD (tail-aware KL in a label-free setting) mechanism. Adding dialogue or summarization SFT experiments would primarily change the surface domain, not the underlying distillation objective, and we expect the relative trends (TAD vs. Vanilla KD) to transfer. Due to space and compute constraints, we prioritized a broad set of pretraining and math-focused supervised experiments that directly probe tail behavior and generalization. Exploring richer SFT settings is an interesting direction for future work, but we believe the current evaluations are sufficient to demonstrate the core advantages of TAD.
>
>
> 3. We added a few code samples in the supplementary material.

---

### Official Review · Reviewer_Ao84 · 2025-10-30

**Soundness:** 3
**Presentation:** 3
**Contribution:** 2
**Rating:** 4
**Confidence:** 4

**Summary:**

This paper introduces Tail-Aware Distillation (TAD), a new method to improve language model distillation. It solves a key problem in standard distillation (using KL divergence), where student models over-focus on the teacher's most likely predictions (the "head") and ignore the informative, low-probability "tail."

TAD uses a new loss function that decouples the head and tail, amplifying the learning signal from the tail. This forces the student to learn the teacher's full knowledge distribution.

Experiments show TAD outperforms other methods in both pre-training and supervised distillation (especially for math) while remaining highly efficient, adding almost no computational cost over standard distillation.

**Strengths:**

1. This paper is clearly written and easy to follow.
2. TAD is more efficient and more effective than the baselines.

**Weaknesses:**

The paper's primary weaknesses are in its experimental design, specifically regarding fair cost comparison and the scale of pre-training.

1. Unfair Computational Budget: The comparison between TAD and the CE (no-KD) baseline is misleading. It ignores TAD's massive upfront computation, namely the teacher inference cost (to generate logits) and the full logit storage cost (making Top-K optimization impossible). A fair comparison would fix the total computational budget, which would require training TAD for significantly fewer steps than the CE baseline.

2. Insufficient Pre-training Scale: Pre-training on only 5B tokens is too small to draw reliable conclusions. The paper would be much stronger if it demonstrated the method's effectiveness at a larger scale, ideally by plotting the scaling law curves for each method (TAD, Vanilla KD, and CE) to show how performance scales with increased compute.

**Questions:**

N/A

---

> ### Author Response · Authors · 2025-12-03
> **Rebuttal**
>
> **Weakness:**
>
> 1. We agree that KD methods incur extra teacher-inference cost compared to pure CE training, but this cost is not specific to TAD. TAD caches only the teacher logits, exactly as in Vanilla KD; computing the top-K subset is done from the already available logits and does not require additional forward passes. We do not store “full logits for all steps” beyond what standard KD practice already assumes.
>
> To address the concern about a fair compute comparison, we also ran pretraining without distillation under a compute-matched budget, matched to the FLOPs of Vanilla KD/TAD runs, and report this computation-matched CLM baseline in Table 4. Even with matched compute, both Vanilla KD and TAD outperform pretraining with only CE loss. For example, when distilling Phi-2 into a 1B model, the FLOP-matched CLM baseline achieves an average score of 46.5 across all tasks, whereas Vanilla KD reaches 49.1 and TAD reaches 50.3. This shows that, under comparable compute, distillation consistently outperforms pretraining with only a CE loss, and the gains are not merely an artifact of extra training steps.
>
> 2. We agree that understanding behaviour at larger pre-training scales is important. To partially address this within our computational budget, we complemented the 2B-token experiments with a scaling-law analysis: we fit a Chinchilla-style scaling curve to the validation loss $\mathcal{L}\_{\mathrm{CLM}}(\mathcal{P}^S) + \mathcal{D}\_{\mathrm{KL}}(\mathcal{P}^T, \mathcal{P}^S)$ for each distillation method (Vanilla KD with forward KL, Vanilla KD with reverse KL, and TAD) on the RegMix validation set, and use it to *predict* performance at larger token budgets. The resulting predictions, reported in Table 5, show that TAD is expected to achieve consistently lower loss than both forward- and reverse-KL Vanilla KD even when extrapolated to substantially larger compute (up to 1T tokens in our analysis), suggesting that the relative gains of TAD are not an artefact of the 5B-token regime.
>
> For the CE baseline with any supervision from the teacher, the composite loss $\mathcal{L}\_{\mathrm{CLM}} + \mathcal{D}\_{\mathrm{KL}}$ used in our scaling-law analysis is not directly applicable. However, for a detailed analysis of the projected validation loss of pretraining with the CE loss versus Vanilla KD, please refer to the MiniPLM paper [4]. We do not repeat the same experiments.
>
> [4] MiniPLM: Knowledge Distillation for Pre-Training Language Models: https://arxiv.org/abs/2410.17215

---

### Official Review · Reviewer_w25M · 2025-10-31

**Soundness:** 3
**Presentation:** 3
**Contribution:** 3
**Rating:** 4
**Confidence:** 5

**Summary:**

For Distillation of LLMs, this paper proposes a modification of KL Divergence loss that decouples and upweighing the loss on the tail of the distribution from that of the head. The proposed loss modification does not require any overhead, and is simple to implement. Across a range of student model sizes (0.5B to 2B) and teacher model sizes (1.5B to 9B), the authors demonstrate consistent improvement across multiple datasets for pre-training, continued pre-training, and fine-tuning when training for short horizons (2B tokens).

**Strengths:**

1. The proposed method is simple and intuitive, and the proposed modification to KLD divergence can be implemented with negligible overhead.
1. The paper is well written and easy to follow, the derivations and proofs are detailed.
1. The authors compare a range of model families and sizes, for both teacher and student, including randomly initialized, pre-trained, etc.
1. The proposed method shows consistent improvement in pre-training, in continued pre-training, and in fine-tuning, particularly for fine-tuning.

**Weaknesses:**

1. The sequence-level normalization in Equation 2 seems somewhat ad-hoc and less principled.
1. The pretraining experiments are extremely extremely small - eg 2B tokens for 1.2B model. This is extremely, extremely far from model convergence (significantly smaller than even chinchilla's "optimal" of ~20x tokens to params), and performance may converge/fall below other methods in longer horizons.
1. Results on most datasets in Table 1/Table 4 are almost at the random value of these datasets (due to the extremely small pre-training duration) - this will somewhat limit the reliability of these scores.
1. The authors method specifically upweighs the tail of a distribution, which has been shown to negatively impact model calibration and downstream fine-tuning.
Given the compute limitations, the paper would perhaps be better positioned if its focus was primarily on the supervised fine-tuning domain, with more extensive SFT experiments. My rating for this paper is closer to 5, but ICLR does not allow that as a score.

**Questions:**

1. Line 87 the authors state "This normalization makes the loss stable for nominal values of $\beta$ such as 1 or 2". But $\beta=1$ corresponds to vanilla KLD - did the authors observe some instability without this modification even for $\beta=1$ ?
1. For Vanilla KD results, I assume the authors used only the KLD loss, without adding the CLM loss? Adding a CLM loss has been shown in multiple prior works to increase KD performance. Can the authors report results with adding the same weightage of CLM loss to KLD as used in their own method, specifically perhaps for Table 6? This will help better isolate effects of the proposed method to that of simply adding CLM loss.
1. Similar to Anshumann et al. (2025), can the authors compare the Expected Calibration Error of the models trained with their method to that of Vanilla KD and to that of just CLM? Anshumann et al. (2025) observed that up-weighing the tail of the distribution improved pre-training scores, but worsens model performance post fine-tuning due to worse calibration of the student - something I conjecture may also happen with the author's method. Specifically, the total probability mass in the student head may be less than the accuracy of the head tokens, and in the tail may be greater than accuracy.
1. This calibration error can also been from the gradient analysis in Equation 5. In general, the teacher will have larger capacity than student, and hence higher accuracy. As LLMs are often well-calibrated, this means $ \sum(p^T_k) > \sum(p^S_k)$ i.e., the teacher will have more probability mass in the head. Let us assume $\beta(X)=2, \sum(p^T_k)=0.8,  \sum(p^S_k)=0.6$ .Then Equation 5 becomes $ \frac{\partial{L}}{\partial{z_i}}= 1.5p^S_i - 2p^T_i $. So even if $p^S_i$ becomes equal to $p^T_i$, this gradient will cause $p^S_i$ to still increase further. Taking any of their trained model, can the authors compare the gradient direction for the tail tokens compared to that of a vanilla KLD?
1. Line 344 the authors state "Adding the cosine loss on hidden states improved both Vanilla KD and TAD". Do the "Vanilla KD" rows in Table 4 include the cosine loss? Can the authors also share results without this cosine loss?
1. (Minor) The authors state "The code is currently not released to preserve anonymity" - the authors could anonymously share the code.
1. (Minor Typo) Line 852 $p^T$ should be $p^S$

---

> ### Author Response · Authors · 2025-12-03
> **Rebuttal**
>
> **Weakness:**
>
> 1. We do not view the sequence-level normalization in Eq.(2) as ad hoc. Let $\alpha_K^T(t)$ denote the teacher’s tail mass for a given sequence $t$, and let $\bar{\alpha}_K^T$ be the average across the sequence. The normalization term in Eq. (2) simply rescales the tail KL by a *scalar function of the sequence*,
> $\beta(X) = \beta \frac{\alpha_K^T(t)}{\bar{\alpha}_K^T},$
> so that sequences where the teacher allocates unusually small or large tail mass do not produce vanishing or exploding tail gradients. In other words, we keep the *shape* of the tail distribution (via KL between the normalized tails) and only adjust the overall scale to stabilize optimization across heterogeneous contexts.
>
> This kind of sequence-level reweighting is not without precedent: sequence-level loss smoothing has been used in prior work such as Elbayad et al. [1], where a sequence-level factor is used to construct a smoothed, reward-based target distribution over alternative outputs. While our formulation is different---we remain fully KL-compatible with the teacher and do not construct new label distributions---the idea of applying a sequence-level normalization to control the contribution of different examples is conceptually similar and principled rather than ad-hoc.
>
> [1] Token-level and sequence-level loss smoothing for RNN language models, https://aclanthology.org/P18-1195/
>
> 2. Chinchilla-style pretraining laws are derived for pretraining from near *random* weights, not for distillation. Even when we use only a causal LM loss, our students are initialized by copying the teacher’s weights in the attention blocks and other layers, which provides a substantial head start compared to regular pretraining and violates the assumptions under which the original Chinchilla analysis was obtained. Moreover, modern pretraining uses different data (code, math, instructions, synthetic) than Chinchilla and improved architectures (RMSNorm, SwiGLU, RoPE, tokenizers), so the Chinchilla token–parameter ratio should be a guideline, not a strict rule.
>
> 3. In Table 1 and Table 4, we deliberately study a from-scratch regime with only 2B tokens, where all methods (Vanilla KD, TAD, MiniPLM, etc.) operate at relatively low absolute performance. Table 1 compares students distilled from Qwen1.5–1.8B against the same architectures used in MiniPLM on the same few-shot tasks, to benchmark against prior work, and Table 4 extends this analysis to larger teachers. The goal of these tables is not to compete with fully trained large models, but to compare *relative* distillation behavior under a fixed, compute-constrained setting. Even in this low-budget regime, TAD consistently outperforms Vanilla KD and MiniPLM on the same tasks and data, indicating that the choice of divergence materially affects sample efficiency. Further, we present the predicted scaling loss in Table 5, indicating that TAD will achieve a lower loss than other methods when trained with a large token budget.
>
> To address reliability at higher performance levels, we complement these results with a continued-pretraining setup (Table 6), where we start from an already pretrained student (TinyLlama-1.1B, trained on 1T tokens) and distill from a powerful teacher (Phi-3). In this setting, the distilled students are far from random (e.g., ARC-C scores are $\ge 32$ across all TAD students) and outperform TinyLlama's 2T checkpoints on all few-shot tasks, demonstrating the efficacy of distillation over additional pretraining for continued learning.
>
> 4. We appreciate the reviewer’s concern about calibration when up-weighting the tail. In the revised version, we added a dedicated section (Section 3.2.3) that evaluates calibration using the Full-ECE metric [2], which is designed for large-vocabulary LMs. Across all teacher–student pairs, we find that TAD is at least as well calibrated as Vanilla KD, and often slightly better calibrated, whereas Reverse-KL exhibits the worst calibration. Thus, in our setting, we do not observe the degradation reported in prior work on tail up-weighting.
>
> Regarding the suggestion to focus primarily on supervised fine-tuning, we agree that SFT is an important application domain and therefore included a supervised distillation setup in Section 3.4, where we further distill the pretrained models (through TAD etc.) using a dataset of GPT-generated solutions for mathematical reasoning tasks, with the results listed in Table 8. In this regime, TAD yields strong downstream performance (e.g., GSM8K scores of **36.8** for TinyLlama-1.1B and **56.0** for Llama2-7B), with no indication of harm from miscalibration. Due to page limitations, we cannot exhaustively cover all SFT settings, but we believe the combination of large-scale pretraining distillation and representative supervised experiments provides a balanced and practically relevant evaluation of TAD.
>
> [2] Full-ECE: A Metric For Token-level Calibration on Large Language Models, https://arxiv.org/pdf/2511.10643

---

> ### Author Response · Authors · 2025-12-03
> **Rebuttal (Continued)**
>
> **Questions:**
>
> 1. Without sequence-level normalization, the loss is given by $\mathcal{D}\_{\mathrm{KL}_1} + \beta \alpha\_K^T \mathcal{D}\_{\mathrm{KL}_2}$. Setting $\beta = 1$ in this expression recovers vanilla KD. However, in this unnormalized form, making the tail divergence $\mathcal{D}\_{\mathrm{KL}_2}$ contribute meaningfully requires choosing $\beta$ substantially larger than 1. In our experiments, a large $\beta$ led to unstable and highly variable gradients across sequences as a function of their tail mass $\alpha\_K^T$.
>
> The normalized formulation in Eq. 2 (Eq. 3 in the revised version) removes the direct dependence on the raw $\alpha\_K^T$, so that $\beta$ directly controls the relative weight of the tail term. Under this parameterization, $\beta = 1$ or $2$ already yields a non-negligible and stable tail contribution, and the resulting loss is meaningfully different from vanilla KD. All distillation experiments in the paper are conducted with this normalized loss.
>
>
> 2. We incorporate both the CLM loss and the KL divergence for Vanilla KD. For TAD, the KL divergence in Eq. 1 is replaced by our tail-aware divergence. We clarified it in Section 2 in the new draft. All the distillation benchmarks, e.g., Vanilla KD, MiniPLM, or Vanilla KD with Reverse KL divergence, include the CLM loss.
>
>
> 3. We have already evaluated the calibration of our models. Across all teacher–student pairs, TAD consistently achieves slightly **lower** Full-ECE than Vanilla KD, while Reverse-KL exhibits the worst calibration. Thus, we do not observe the degradation reported by Anshumann et al. Moreover, in our supervised distillation experiments for mathematical reasoning (Section~3.4), TAD students achieve strong post–fine-tuning performance (e.g., GSM8K scores of 36.8 for TinyLlama-1.1B and 56.0 for Llama2-7B), with no evidence of degradation due to miscalibration. Taken together, neither the Full-ECE measurements nor the supervised results support the conjecture that TAD worsens calibration relative to Vanilla KD or CLM under the same compute budget.
>
>
> 4. The increase in head or tail mass depends less on one particular token. Let us denote the head/tail index sets by $\mathcal{H}$ and $\mathcal{T}$. As per the assumption $\sum_{h\in\mathcal{H}}p^T_h=0.8$, $\sum_{h\in\mathcal{H}}p^S_h=0.6$, and $\beta(X)=2$. Let $p^S=\mathrm{softmax}(z)$ be the student distribution, $p^T$ the teacher. Therefore, $\sum_{t\in\mathcal{T}}p_t^T=0.2$ and $\sum_{t\in\mathcal{T}}p_t^S=0.4$. Consequently, it is not possible for all $p^T_t = p^S_t$ for the tail tokens ($t \in \mathcal{T}$).
>
> From Eq.(5) of the paper, for $\beta(X)=2$ and this mass configuration, the per-logit gradients are (since the head gradient is the same as Vanilla KL):
>
> $$\forall t\in\mathcal{T}:\quad g_t = 1.5p_t^S - 2p_t^T,
> \qquad
> \forall h\in\mathcal{H}:\quad g_h = p_h^S - p_h^T.$$
>
> Using the softmax Jacobian identity $\frac{\partial p_i}{\partial z_j}=p_i(\delta_{ij}-p_j)$, the first-order change of **head** mass $P_{\text{head}}^S=\sum_{h\in\mathcal{H}}p_h^S$ under a small gradient-descent step $z\leftarrow z-\eta g$ is:
>
> $$\Delta P^S_{\text{head}} \approx -\eta\left(\sum_{h\in\mathcal{H}} p_h^S g_h - 0.6\sum_{j} p_j^S g_j\right) = \eta\left[0.6\sum_{t\in\mathcal{T}} p_t^S(1.5p_t^S-2p_t^T) - 0.4\sum_{h\in\mathcal{H}} p_h^S(p_h^S-p_h^T)\right].$$
>
> Therefore, the *necessary and sufficient* (first-order) condition for head mass to *increase* is:
>
> $$\Delta P^S_{\text{head}} > 0
> \Longleftrightarrow
> 0.6\sum_{t\in\mathcal{T}} p_t^S\bigl(1.5p_t^S-2p_t^T\bigr)>0.4\sum_{h\in\mathcal{H}} p_h^S\bigl(p_h^S-p_h^T\bigr).$$
>
> Since total probability is conserved, $\Delta P^S_{\text{tail}} = -\Delta P^S_{\text{head}}$. Thus the equation above simultaneously implies $\Delta P^S_{\text{tail}}<0$.
>
> Let us assume a simple case in which the teacher and the student have similar *shapes* in the head and tail, differing mainly in total mass:
>
> $$\forall h\in\mathcal{H}:\quad p_h^T=\tfrac{0.8}{0.6}p_h^S \quad\text{and}\quad
> \forall t\in\mathcal{T}:\quad p_t^T=\tfrac{0.2}{0.4}p_t^S .$$
>
> Then:
>
> $$\sum_{h\in\mathcal{H}} p_h^S(p_h^S-p_h^T)
> =\Bigl(1-\tfrac{4}{3}\Bigr)\sum_{h\in\mathcal{H}}(p_h^S)^2
> =-\tfrac{1}{3}\sum_{h\in\mathcal{H}}(p_h^S)^2,
> \qquad
> \sum_{t\in\mathcal{T}} p_t^S(1.5p_t^S-2p_t^T)
> =\tfrac{1}{2}\sum_{t\in\mathcal{T}}(p_t^S)^2,$$
>
> so the bracket in the equation for $\Delta P^S_{\text{head}}$ becomes:
>
> $$0.6\cdot\tfrac{1}{2}\sum_{t\in\mathcal{T}}(p_t^S)^2
> -0.4\cdot\Bigl(-\tfrac{1}{3}\sum_{h\in\mathcal{H}}(p_h^S)^2\Bigr)
> = 0.3\sum_{t\in\mathcal{T}}(p_t^S)^2 + \tfrac{2}{15}\sum_{h\in\mathcal{H}}(p_h^S)^2 > 0.$$
>
> Hence, in this setting with $\sum_{h\in\mathcal{H}}p_h^T=0.8$, $\sum_{h\in\mathcal{H}}p_h^S=0.6$, $\beta(X)=2$, we have:
>
> $$\boxed{\ \Delta P^S_{\text{head}} > 0 \quad\text{and}\quad \Delta P^S_{\text{tail}} < 0\ }.$$
>
> Thus, the student's head mass $P_{\text{head}}^S$ continues to increase when $\sum_{h\in\mathcal{H}}p^T_h=0.8$ and $\sum_{h\in\mathcal{H}}p^S_h=0.6$.

---

> ### Author Response · Authors · 2025-12-04
> **Rebuttal (Continued)**
>
> **Questions**
>
> 5. In Table 4 (Section 3.2.2), all methods---including Vanilla KD, TAD, and Reverse-KL---are trained with an additional cosine loss on the hidden states. We use this auxiliary loss only in the very low-token, from-scratch regime (2B tokens) to stabilize training and make the comparison between divergences more informative. Without it, all methods perform considerably worse, and the absolute results become even closer to random. For experiments *without* any hidden-layer loss, please refer to Table 1, where we report results for Vanilla KD and TAD using only the KL-based objectives.
>
> 6. We released a part of the code
>
> 7. Corrected it

---

### Official Review · Reviewer_6wm4 · 2025-11-03

**Soundness:** 3
**Presentation:** 2
**Contribution:** 2
**Rating:** 2
**Confidence:** 3

**Summary:**

This paper proposes Tail-Aware Distillation (TAD), a novel knowledge distillation method for training smaller causal language models from larger teachers. Its key contribution is decoupling the KL divergence loss into a top-K term and a tail term, amplifying the gradient contribution of the tail probabilities to prevent the student from over-focusing on the most likely tokens. The method is computationally efficient, requiring similar FLOPs to vanilla KD, and enables effective large-scale pretraining and supervised distillation even with limited resources. Extensive experiments show TAD produces competitive or superior students across various model sizes and tasks, including mathematical reasoning, outperforming recent methods like MiniPLM.

**Strengths:**

This paper's primary strengths are threefold.

First, it presents a novel and principled adaptation of Decoupled Knowledge Distillation, transitioning it from supervised image classification to the label-free domain of language model pre-training via a new rank-based (Top-K vs. tail) decoupling.

Second, it is supported by rigorous theoretical foundations, including a detailed derivation of the proposed loss and an insightful gradient analysis that explains the method's dynamics.

Finally, the work is substantiated by comprehensive experiments demonstrating the algorithm's effectiveness and versatility across diverse settings—including pre-training distillation from scratch, domain adaptation for specialized skills like mathematical reasoning, and supervised fine-tuning—while consistently maintaining a computational efficiency comparable to the highly practical Vanilla Knowledge Distillation.

**Weaknesses:**

1. The introduction fails to clearly articulate the motivation behind the proposed method. The rationale for developing Tail-Aware Distillation (TAD) only becomes apparent later in Section 2, which disrupts the logical flow and makes it difficult for readers to grasp the core contribution early on.

The formalization of the method is not self-contained. For instance, the notation \mathcal{P} is used without a clear definition, and the formulation of vanilla knowledge distillation is not properly introduced. This lack of clarity hinders understanding, especially for readers unfamiliar with the baseline approaches.

2. The authors explicitly state that TAD is not a variant of Decoupled Knowledge Distillation (DKD), arguing that DKD is label-anchored while TAD is rank-anchored. However, when K = 1 and the teacher's top-1 token is treated as a pseudo-label, the two methods exhibit substantial similarities. This suggests that TAD can be viewed as an adaptation of DKD to the language modeling domain, rather than a fundamentally new approach. The distinction drawn by the authors appears overstated and may misrepresent the true nature of their contribution.

3. The method introduces two new critical hyperparameters: the Top-K value and the tail weight coefficient \beta. Although a sensitivity analysis is provided, showing robustness around K = 5-10 and \beta = 2, this adds non-trivial complexity and tuning cost compared to Vanilla KD, which has no such parameters. For practitioners, this necessitates extra work to determine the optimal configuration for different teacher-student pairs and data domains, potentially hindering the method's ease of adoption.

4. The experimental comparisons are primarily limited to Vanilla KD, Sequence-KD, MiniLLM, and MiniPLM. The evaluation would be more comprehensive and convincing if it included other relevant and advanced distillation techniques. For instance, comparisons with methods using Reverse KL divergence (On-policy KD) are absent. Including these baselines would better situate TAD's performance within the broader landscape of knowledge distillation research and provide a clearer understanding of its relative advantages.

**Questions:**

Given that the core idea of decoupling the KL divergence into target vs. non-target (or top-K vs. tail) components was originally proposed in Decoupled Knowledge Distillation (DKD, Zhao et al., 2022) for vision tasks, could the authors more explicitly clarify the novelty of their method beyond a direct adaptation of DKD to the language modeling domain?

While the shift from image classification to causal LM pretraining is non-trivial, the fundamental decomposition and re-weighting strategy appear conceptually similar. What are the key algorithmic or theoretical innovations in TAD that distinguish it from a straightforward application or extension of DKD to language?

---

> ### Author Response · Authors · 2025-12-03
> **Rebuttal**
>
> **Weakness:**
>
> 1. To address the formalization issue, we now explicitly define the prediction space $\mathcal{P}$ as the simplex of token distributions produced by a model (e.g., $\mathcal{P}(S)$ for the student and $\mathcal{P}(T)$ for the teacher), and we first introduce vanilla KD as forward KL on $\mathcal{P}(T)$ and $\mathcal{P}(S)$ before presenting TAD. This makes Section~2 self-contained for readers who are less familiar with standard KD. Please refer to the revised manuscript. Given the strict page limit, the methodology must remain concise, but we have streamlined it so that the motivation and core idea of TAD are stated earlier.
>
> 2. As we stated, TAD is inspired by DKD, and that, when $K=1$, there is a superficial similarity in ``decoupling" a head term from a residual term. However, TAD is not simply DKD with a different name. As discussed in Section-2 and illustrated in Figure-2, DKD is inherently **label-anchored**: it requires a target vs. non-target split based on ground-truth or pseudo-labels. In our unsupervised LM pretraining setting, the teacher’s argmax and the next-token label disagree on roughly 39% to 46% of the positions in a generic corpus, so treating the teacher’s top-1 token as a pseudo-label would systematically inject incorrect supervision and make a direct DKD-style formulation impractical. This is where we introduce TAD, a **fully label-free** and **rank-based** method that operates on the teacher and student distributions over the entire vocabulary, splitting them into top-$K$ and tail by rank. It is a distinct objective specifically designed to be workable in the causal LM distillation regime where reliable labels (or pseudo-labels) are not available.
>
> 3. Even if our algorithm has more hyperparameters, that does not mean it should be difficult to fine-tune. We designed the loss so that $\beta = 1$ or $2$ produces the best result. We showed that the sensitivity across $\beta$ is minimal in Table 2. Further, we explained why we do not need to extend $K$ beyond 20. Thus, TAD can be deployed with a single default such as $K \in {5,10}$ and $\beta = 2$, which we have found to work well in all settings, thus limiting the need for extensive hyperparameter tuning.
>
> 4. We already included the KD results using Reverse KL in Table 4. We included Vanilla KD, Sequence-KD, MiniLLM, and MiniPLM as our baselines, with MiniPLM being the most recent. These constitute an exhaustive set for our experiments, and our results are more extensive than those in most ICLR papers. On-policy KD is primarily defined in supervised settings (datasets containing both $x$ and $y$). In contrast, we focus on pretraining distillation in our experiments (only $x$), and therefore, On-Policy KD is not directly applicable in our case.
>
> **Questions:**
>
> We already explained the novelty of TAD with respect to DKD in section 2 of the paper, and later explained in response to weakness 2. Even if TAD is inspired by DKD through the idea of decoupling head vs. tail mass, it introduces several algorithmic and theoretical innovations that go beyond a straightforward DKD-style extension to language. First, TAD is explicitly designed for **label-free** causal LM pretraining: the loss is defined purely in terms of the teacher’s ranked probabilities over the vocabulary, without ever conditioning on ground-truth or pseudo-labels. This is essential in generic corpora where the teacher argmax and the next-token label disagree in 39% to 46% of positions, making a DKD-style target/non-target split unreliable.
>
> Second, TAD does not simply reweight non-target probabilities as in DKD: it also **renormalizes** the tail KL term using the teacher’s average tail mass over the sequence, which stabilizes gradients across heterogeneous contexts and makes a single global $\beta$ viable in practice.
>
> Finally, we provide a gradient-level analysis showing that this construction preserves the teacher distribution as a fixed point of the student, while selectively amplifying tail corrections only when the student over-concentrates head mass—a behavior that does not follow from the original DKD formulation.

---

### Author Response · Authors · 2025-12-04
**Summary of Rebuttal**

We summarize the key changes in the revised submission. All the changes in the primary draft are highlighted in a blue font.

1. We clarified the notation (e.g., $\mathcal{P}$) and explicitly introduced the Vanilla KD loss in Section 2 as $\mathcal{L}\_{\mathrm{CLM}}(\mathcal{P}^S) + \mathcal{D}\_{\mathrm{KL}}(\mathcal{P}^T, \mathcal{P}^S)$, following the request from reviewer 6wm4.

2. We add a CLM-only pretraining baseline with a *compute-matched* budget, matched to the FLOPs of Vanilla KD/TAD in Table 4, following the concern from the reviewer Ao84 regarding a fair computational budget for all baselines.

3. We report the calibration error using Full-ECE for the experiments in Table 4 and add Section 3.2.3 to explain the motivation of using Full-ECE instead of top-1-based ECE, following the question from reviewer w25M on the calibration error of the distilled models.

4. We include a scaling-law analysis in Table 5, predicting validation loss when scaling up to 1T tokens, again following a request from reviewer Ao84.

5. We add new experiments in Table 6 on continued pretraining of TinyLlama-1.1B via distillation, which show substantially improved results when starting from an already pretrained model, in comparison to training from scratch as in Tables 1 and 4, addressing the weakness raised by reviewer w25M.

6. We released a few sample codes as the supplementary material, addressing the concerns regarding reproducibility raised by reviewer w25M and gFdZ

---

### Public Comment · ~Sayantan_Dasgupta1 · 2026-05-02
**Accepted to ICML 2026**

This work has been accepted to ICML 2026

---

### Meta-Review · Area_Chair_JEwk · 2026-01-06

**Summary:**

The paper introduces Tail-Aware Distillation (TAD), a method that modifies the standard KL divergence loss in knowledge distillation (KD). TAD calculates the KL divergence only over non-top-K tokens and scales this loss by the residual of the top-K token probabilities. This approach is intended to explicitly incorporate "tail information," leading to improved KD performance.

Reviewers raised several critical points regarding the proposed method:
- Marginal Novelty: TAD is seen as having only minor differences from existing methods like DKD, primarily in its rank-based approach with a general K versus DKD's label-based approach.
- Insufficient Theoretical Justification: The concept of "tail amplification" lacks theoretical underpinning, specifically concerning its impact on convergence and stability.
- Computational Overhead: The processes of identifying top-K tokens, scaling, and using position information introduce extra computational cost, leading to questions about whether comparisons with CE-only baselines were conducted under compute-matched conditions.

**Reviewer Concerns:**

The authors provided the following clarifications:
- Explicit Design and Stabilization: TAD is specifically designed for label-free causal language model pretraining. A key stabilizing element is the renormalization of the tail KL term using the sequence-level teacher tail mass, which helps stabilize gradients.
- Gradient Analysis: Analysis at the gradient level was provided, which suggests that the teacher distribution remains the unique fixed point. This analysis also indicates that tail amplification primarily acts to counteract overly confident student heads rather than injecting arbitrary noise.
- Compute-Matched Baselines: The authors added compute-matched CE baselines. These results show that both Vanilla KD and TAD surpass CE-only training when matched for FLOPs.

Despite the authors' efforts, certain concerns persist:
- Novelty Remains a Concern: While the authors explained the distinction of the loss, the perceived similarity to existing work makes the novelty issue a reasonable and major concern for reviewers. Although the loss itself might not be a critical reason for rejection, given that many papers introduce variations of existing losses, it must be thoroughly justified. Reviewers felt that the empirical justification and discussion of pros and cons were insufficient given the marginal novelty.
- Missing Theoretical Justification: The lack of theoretical justification for the approach is still a significant concern, as the empirical evidence and discussion were insufficient.
- Need for Fair Comparison: Including wall-clock time information would significantly help in convincing reviewers of the fairness of the computational comparisons.

**Reviewer Scores:**

Some reviewers may raise their scores slightly, but no substantial change is expected. The anticipated ratings are 4, 4, 6, and 6, yielding an average score of 5.

---

### Decision · Program_Chairs · 2026-01-26

Reject